# KokerNet: Koopman Kernel Network for Time Series Forecasting

## Abstract

The Koopman operator has gained increasing attention in time series forecasting due to its ability to simplify the complex evolution of dynamic systems. However, most existing Koopman-based methods suffer from significant computational costs in constructing measurement functions and struggle to address the challenge posed by the variation in data distribution. Additionally, these approaches tend to empirically decompose time series or distributions into combinations of components, lacking interpretability. To tackle these issues, we propose a novel approach, **Ko**opman **ker**nel **net**work (**KokerNet**), for time series forecasting. On one hand, we construct a measurement function space using the spectral kernel method, which enables us to perform Koopman operator learning in a low-dimensional feature space, efficiently reducing computational costs. On the other hand, an index is designed to characterize the stationarity of data in both time and frequency domains. This index can interpretably guide us to decompose the time series into stationary and non-stationary components. The global and local Koopman operators are then learned within the constructed measurement function space to predict the future behavior of the stationary and non-stationary components, respectively. Particularly, to address the challenge posed by the variation in distribution, we incorporate a distribution module for the non-stationary component, ensuring that the model can make aligned distribution predictions. Extensive experiments across multiple benchmarks illustrate the superiority of our proposed KokerNet, consistently outperforming the state-of-the-art models.

## 1 Introduction

Time series forecasting has long been a focus of attention in various real-world applications, such as the forecasting of weather (Wu et al., 2023b), traffic (Jiang et al., 2023), and disease propagation (Matsubara et al., 2014). Over the past decades, researchers have developed a range of time series forecasting models, such as RNN-based models (Qin et al., 2017; Lai et al., 2018; Jia et al., 2023; Schirmer et al., 2022), Transformer-based models (Li et al., 2019; Zhou et al., 2021; Wu et al., 2021; Zhang & Yan, 2022; Zhou et al., 2022; Liu et al., 2021; Nie et al., 2022; Wen et al., 2023; Liu et al., 2024), TCN-based models (Wu et al., 2023a), and MLP-based models (Zeng et al., 2023; Challu et al., 2023; Yi et al., 2023; Oreshkin et al., 2020).

Recently, Koopman operator (Koopman, 1931) has gained increasing attention in the time series forecasting task since it can simplify the intricate modeling process of dynamic system evolution, by acting on measurement functions. From one perspective, some studies determine the measurement functions through dynamic mode decomposition (DMD) and further learn the Koopman operator to describe the evolution of the time series (Wang et al., 2023a;b). However, the use of singular value decomposition (SVD) in DMD results in significant computational costs, particularly in high-dimensional spaces. From another perspective, several methods employ the decomposition strategy to divide the time series or data distribution into combinations of components and further learn specialized Koopman operators for different components (Zhang et al., 2024; Liu et al., 2023). These approaches evade information loss from the single-component assumption. However, these decompositions often rely on empirical determinations of component composition and proportions, lacking interpretability. In addition, these methods share a common limitation in that they do not consider the problem of prediction accuracy caused by distribution changes.

To address these issues, we propose a novel time series forecasting method, namely, **Ko**opman **ker**nel **net**work (**KokerNet**). Concretely, we first construct a measurement function space in the form of a reproducing kernel Hilbert space (RKHS) spanned by cosine functions. This scheme enables us to learn the Koopman operator in a low-dimensional feature space, significantly reducing the computational cost. Next, an index $S_v$ is designed by performing the Kolmogorov-Smirnov (KS) test on both the data and spectrum aspects, which interpretably guides us in decomposing the time series into stationary and non-stationary components. Based on the decomposition, the global shared and local operators are learned within the constructed RKHS to predict the stationary and non-stationary components, respectively. Moreover, to tackle the challenges posed by the time-varying distribution, we incorporate a distribution constraint module into the forecasting process. This inclusion ensures that the forecasts align with the actual distribution.

The main contributions of this paper are shown as follows:

- We model the evolution of the time series as temporal dependence using the spectral kernel method, naturally resulting in a measurement function space spanned by a set of cosine functions. Compared to the Koopman-based studies that determine the measurement function via DMD, KokerNet allows us to learn the Koopman operators in a low-dimensional feature space, significantly reducing the computational cost. Moreover, in the constructed space, the measurement functions are higher-order derivatives, and any derivative of a function is itself a composition of the function. This enables us to supervise any derivative of the measurement function with complicated time series.

- Following the general framework (Liu et al., 2023), KokerNet decomposes the time series into stationary and non-stationary components. The global and local Koopman operators are then optimized to capture the dynamics of each component. Notably, rather than relying on empirical decomposition, we design an index $S_v$ based on the KS test to provide an interpretable guide for the decomposition.

- To tackle the common limitation of the Koopman-based approaches posed by the time-varying distribution, we incorporate a distribution constraint into the forecasting process, ensuring that the prediction aligns with the temporal variation in distribution. Non-stationary time series can be viewed as discrete signals where unobserved parts differ significantly from the observations, motivating us to consider the distribution of non-stationary components.

- We conduct extensive experiments. The results demonstrate that our approach is superior to state-of-the-art models. In addition, we explore the influence of the designed index on the decomposition of the time series.

## 2 PRELIMINARIES

### 2.1 NOTATION

Formally, we use $\mathbb{R}^n$ and $\mathbb{R}^{m \times n}$ to denote $n$-dimensional Euclidean spaces and the space of $m \times n$ real-valued matrix. Throughout the paper, the matrices, vectors, and scalars are denoted by bold capital letters (*e.g.* $\boldsymbol{X}$), bold lower-case letters (*e.g.* $\boldsymbol{x}$) and lower-case letters (*e.g.* $x$), respectively. $\boldsymbol{X}_T = \{\boldsymbol{x}_1, \boldsymbol{x}_2, \ldots, \boldsymbol{x}_T\}$ denotes the time series or trajectory with $T$ time points.

### 2.2 KOOPMAN THEORY

For a complicated dynamical system, its evolution can be formulated as $\boldsymbol{s}_{t+1} = \boldsymbol{F}(\boldsymbol{s}_t)$, where $\boldsymbol{s}_t$ denotes the system state on moment $t$, and $\boldsymbol{F}$ denotes the flow map of transferring the system state on moment $t$ to moment $t + 1$. However, it is a challenge to identify the complex evolution with a flexible but parsimonious $\boldsymbol{F}$. Koopman theory (Koopman, 1931) has been developed to analyze complex dynamic systems. Its core idea is to characterize the complicated evolution via an infinite-dimensional linear Koopman operator. By acting on the measurement function, this operator advances the system as follows:

$$\mathcal{K}f(\boldsymbol{s}_t) = f(\boldsymbol{F}(\boldsymbol{s}_t)) = f(\boldsymbol{s}_{t+1}), \tag{1}$$

where $\mathcal{K}$ is the Koopman operator, $f : \mathbb{R}^d \to \mathbb{R}^D, D \to \infty$ is the measurement function. When $D \to \infty$, it incurs extensive computational costs, which hinders the practical use of this method.

## 3 KOKERNET

In this section, we first present the construction of the measurement function space and the corresponding finite-dimensional Koopman operator, which advances the time series by acting on the constructed function space. Subsequently, we introduce a specially designed index to guide the time series decomposition into its stationary and non-stationary components. Based on this decomposition, the global shared and local Koopman operators are learned in the constructed measurement function space to characterize the dynamics of stationary and non-stationary components, respectively. Finally, we introduce a distribution constraint. The architecture of Koopman operator learning and the distribution constraint is shown in appendix B.

### 3.1 MEASUREMENT FUNCTION SPACE CONSTRUCTION

In this paper, we model the complex evolution $\boldsymbol{x}_t \rightarrow \boldsymbol{x}_{t+1}$ of the time series $\boldsymbol{X}_T = \{\boldsymbol{x}_1, \boldsymbol{x}_2, \ldots, \boldsymbol{x}_T\}$ as its temporal dependence using the kernel method, which has been proved to be a promising approach to simplify the intricate correlation with an implicit feature mapping. Hence, the temporal dependence is formulated as the following kernel function:

$$k(\boldsymbol{x}_t, \boldsymbol{x}_{t+1}) = \langle f(\boldsymbol{x}_t), f(\boldsymbol{x}_{t+1}) \rangle_{\mathcal{H}}, \tag{2}$$

where $f$ is the implicit feature mapping, which can be considered as the measurement function since it has the potential of mapping the data into an infinite-dimensional feature space. $\mathcal{H}$ is the RKHS, induced by the kernel $k(\cdot, \cdot)$.

Based on Bochner's theorem and the following Theorem 3.1, we can approximate the kernel function $k(\cdot, \cdot)$ by a low-dimensional feature mapping $g : \mathbb{R}^d \rightarrow \mathbb{R}^M$, such that:

$$g(\boldsymbol{x}_t) = \frac{2}{\sqrt{M}}[\cos(\boldsymbol{w}_1 \boldsymbol{x}_t + b_1), \cos(\boldsymbol{w}_2 \boldsymbol{x}_t + b_2), \ldots, \cos(\boldsymbol{w}_M \boldsymbol{x}_t + b_M)]^{\top},$$

$$k(\boldsymbol{x}_t, \boldsymbol{x}_{t+1}) \approx \bar{k}(\boldsymbol{x}_t, \boldsymbol{x}_{t+1}) = \langle g(\boldsymbol{x}_t), g(\boldsymbol{x}_{t+1}) \rangle_{\bar{\mathcal{H}}}, \tag{3}$$

where $g \in \bar{\mathcal{H}}$ is the approximation of $f$, $\bar{\mathcal{H}}$ is the constructed measurement function space, which is an RKHS induced by $\bar{k}(\cdot, \cdot)$. The detailed derivation is shown in appendix C.1.

**Definition 3.1.** *We say that a matrix $\boldsymbol{A}$ is a $\Delta$-spectral approximation of another matrix $\boldsymbol{B}$, if $(1 - \Delta)\boldsymbol{B} \preceq \boldsymbol{A} \preceq (1 + \Delta)\boldsymbol{B}$.*

**Theorem 3.1.** *Sample $\boldsymbol{w}_1, \boldsymbol{w}_2, \cdots, \boldsymbol{w}_M$ according to a spectral density function $p(\boldsymbol{w})$ and set $\boldsymbol{Z} = g(\boldsymbol{X}_T)$. When the sampling number $M \geq \frac{2\delta(3\sqrt{n} + 2\Delta)}{3\Delta^2} ln\frac{8\sqrt{n}}{\rho}$, with the probability of at least $1 - \rho$, $\boldsymbol{Z}\boldsymbol{Z}^{\top}$ is the $\Delta$-spectral approximation of $\boldsymbol{K} = \langle f(\boldsymbol{X}_T), f(\boldsymbol{X}_T) \rangle_{\mathcal{H}}$.*

*Proof.* The proof is relegated to the appendix C.2 of our paper due to space limitations. □

Based on the Koopman theory (*i.e.*, eq. (1)), for the measurement function $g \in \bar{\mathcal{H}}$, there exists an finite-dimensional Koopman operator $\bar{\mathcal{K}}$ that:

$$\bar{\mathcal{K}}g(\boldsymbol{x}_t) = g(\boldsymbol{F}(\boldsymbol{x}_t)) = g(\boldsymbol{x}_{t+1}). \tag{4}$$

Furthermore, define

$$g_{\omega}(\boldsymbol{x}) = \frac{1}{T} \int_{\tau=0}^{T} g(\boldsymbol{x}_{\tau}) e^{-i\omega\tau} d\tau, \tag{5}$$

for the finite trajectory $\{\boldsymbol{x}_1, \boldsymbol{x}_2, \ldots, \boldsymbol{x}_T\}$ (*i.e.*, the time series $\boldsymbol{X}_T$) based on $g \in \bar{\mathcal{H}}$.

**Theorem 3.2.** *For every eigenfrequency $\omega \in \mathbb{R}$ of the Koopman operator $\bar{\mathcal{K}}$, Let $g$ be the measurement function on the finite trajectory $\{\boldsymbol{x}_1, \boldsymbol{x}_2, \ldots, \boldsymbol{x}_T\}$. Then,*

*(i) when $T \geq \sqrt{\frac{2}{M}}\frac{3\omega_{max}}{\epsilon}$, $g_{\omega}$ can approximate any Koopman eigenfunction with $\epsilon$ accuracy, for $\epsilon > 0$.*

*(ii) $\lim_{T\to\infty} g_{\omega}$ is an eigenfunction of the Koopman operator $\mathcal{K}$.*

*Proof.* The proof is relegated to appendix C.3 of our paper due to space limitations. □

According to Theorem 3.2, we can observe that $g_\omega$ is the eigenfunction of the Koopman operator $\bar{\mathcal{K}}$, corresponding to the eigenfrequency $\omega$, and when $T \to \infty$, $\bar{\mathcal{K}}$ is the approximation of the infinite-dimensional Koopman operator $\mathcal{K}$. The defination in eq. (5) provides a pleasing scheme, identifying the eigenfunction and eigenfrequency from the data directly, to capture the complex dynamic patterns and simplify the evolution of the time series.

All in all, we construct a measurement function space $\bar{\mathcal{H}}$, which is spanned by a set of cosine functions. The measurement function $g \in \bar{\mathcal{H}}$ enables us to describe the complicated evolution of the time series via a finite-dimensional Koopman operator $\bar{\mathcal{K}}$ directly, such that:

$$\bar{\mathcal{K}}g(\boldsymbol{x}_t) = g(\boldsymbol{F}(\boldsymbol{x}_t)) = g(\boldsymbol{x}_{t+1}). \tag{6}$$

## 3.2 KOOPMAN OPERATOR LEARNING

Once the measurement function space $\bar{\mathcal{H}}$ is constructed, the Koopman operator is learned to describe the evolution of the time series within $\bar{\mathcal{H}}$. The real-world time series typically contains both time-invariant and time-variant patterns, corresponding to stationary and non-stationary components. It is arbitrary to assume the entire time series is stationary or non-stationary without any prior knowledge, as this easily leads to information loss or the introduction of unnecessary disturbances. Assuming the time series is fully stationary would lose the non-stationary information, while assuming the time series is fully non-stationary would introduce uncertainty in the stationary component, resulting in suboptimal predictions. Hence, in this section, we first decompose the time series $\boldsymbol{X}_T$ into stationary and non-stationary components, i.e., $\boldsymbol{X}_T = \boldsymbol{X}_s + \boldsymbol{X}_{ns}$, where $\boldsymbol{X}_s$, $\boldsymbol{X}_{ns}$ denote the stationary and non-stationary components, respectively. Then, the global shared and local Koopman operators are separately learned to capture the evolution of these two components.

**Time Series Decomposition** For the time series decomposition, we designed an index based on the KS test to determine the proportions of each component from both time and frequency domains. It is worth noting that the KS test measures the consistency of distributions between different periods after removing global trends and seasonal effects. These operations would cause the residual of the time series to tend to be stochastic fluctuation, which is the key attribution of the non-stationarity of real-world time series. More precisely, in the time domain, we divide the time series $\boldsymbol{X}_T$ into $J$ segments, i.e., $\boldsymbol{X}_T = [\boldsymbol{x}_1, \boldsymbol{x}_2, \ldots, \boldsymbol{x}_J], \boldsymbol{x}_j, \boldsymbol{x}_{j+1} \in \mathbb{R}^{C \times \frac{T}{J}}, j = 1, 2, \ldots, J-1$ represent adjacent time series segments, and their corresponding detrended and deseasonalized residuals are $\boldsymbol{x}_j^r, \boldsymbol{x}_{j+1}^r \in \mathbb{R}^{C \times \frac{T}{J}}$. The statistical magnitude of KS test is then computed as follows:

$$p = \frac{1}{J-1} \sum_{j=1}^{J-1} p_j, \quad p_j = \frac{1}{C} \sum_{c=1}^{C} sup|G_j(x_c) - G_{j+1}(x_c)|, \tag{7}$$

where $C$ is the number of the variate, $G_j(x_c), G_{j+1}(x_c)$ are the empirical cumulative distribution function for the $c$-th variate of $\boldsymbol{x}_j^r, \boldsymbol{x}_{j+1}^r$, respectively.

For the frequency domain, the Wiener–Khinchin theorem shows that the autocorrelation function and the power spectral density function are a pair of Fourier transforms, such that:

$$R(\tau) = \int_{-\infty}^{\infty} S(\lambda)e^{i2\pi\lambda\tau}d\lambda, \quad S(\lambda) = \int_{-\infty}^{\infty} R(\tau)e^{-i2\pi\lambda\tau}d\tau, \tag{8}$$

where $R(\tau) = \int_{-\infty}^{\infty} \boldsymbol{x}_t \boldsymbol{x}_{t-\tau} dt$ is the autocorrelation function, which measures the relationship between a time series and its lagged versions. we can observe that the stationarity is manifested in the uncertainty about the spectrum $\lambda$. Therefore, similar to the time domain, the statistical magnitude of KS test in the frequency domain is computed by:

$$\bar{p} = \frac{1}{J-1} \sum_{j=1}^{J-1} \bar{p}_j, \quad \bar{p}_j = \frac{1}{\bar{C}} \sum_{\bar{c}=1}^{\bar{C}} sup|G_j(\lambda_{\bar{c}}) - G_{j+1}(\lambda_{\bar{c}})|, \tag{9}$$

where $G_j(\lambda_{\bar{c}}), G_{j+1}(\lambda_{\bar{c}})$ are the empirical cumulative distribution function for the $\bar{c}$-th variate of $\boldsymbol{\lambda}_j, \boldsymbol{\lambda}_{j+1}$, respectively, and $\boldsymbol{\lambda}_j$ is the spectrum, obtained by performing the Fourier transform for the segments $\boldsymbol{X}_T = [\boldsymbol{x}_1, \boldsymbol{x}_2, \ldots, \boldsymbol{x}_J]$.

We define the index $S_v = p\bar{p}$. It captures the evolving patterns in both the time and frequency domains via multiplication. The time series tends to be stationary when the values of $p$ and $\bar{p}$ are

small, whereas higher values indicate non-stationary. Thus, the index $S_v = p\bar{p}$ offers a credible insight into the stationarity of the data and guides us to decompose the data into stationary and non-stationary components. The detailed decomposition process can be found in appendix D.

**Koopman Operator Learning**   After decomposing the time series into stationary and non-stationary components, the global shared and local Koopman operators are respectively learned to describe the evolution of these two components in the constructed measurement function space $\bar{\mathcal{H}}$. For the stationary component, we design a global shared Koopman operator $\mathcal{K}_s$ to capture the consistent variation patterns of the time series. This operator advances the evolution of the time series in the measurement function space, such that:

$$\mathcal{K}_s \boldsymbol{Z}_s^{\text{back}} = \boldsymbol{Z}_s^{\text{fore}}, \quad \boldsymbol{Z}_s^{\text{back}} = g(\boldsymbol{X}_s^{\text{back}}), \quad \boldsymbol{Z}_s^{\text{fore}} = g(\boldsymbol{X}_s^{\text{fore}}), \quad \boldsymbol{X}_s^{\text{fore}} = \Phi_{\text{de}}(\boldsymbol{Z}_s^{\text{fore}}), \qquad (10)$$

where $\boldsymbol{X}_s^{\text{back}}$ is the stationary component of the current time series, and $\boldsymbol{X}_s^{\text{fore}}$ is the stationary component that is going to be predicted. $g \in \bar{\mathcal{H}}$ is the measurement function, acting as on encoder, and $\Phi_{\text{de}}$ denotes the decoder.

For the non-stationary component, dynamic Koopman operator $\mathcal{K}_{\text{ns}}$ are learned to capture the local dynamics of the time series. Concretely, we divide the time-variant component $\boldsymbol{X}_{\text{ns}}$ into $Q$ segments, assuming $T$ is divisible by $Q$. The segmentation is defined as:

$$\boldsymbol{X}_{\text{ns}} = [\boldsymbol{x}_1, \boldsymbol{x}_2, \ldots, \boldsymbol{x}_Q], \quad \boldsymbol{x}_i = [\boldsymbol{x}_{(i-1)\frac{T}{Q}+1}, \ldots, \boldsymbol{x}_{i\frac{T}{Q}}] \in \mathbb{R}^{C \times \frac{T}{Q}}, i = 1, 2, \ldots, Q. \qquad (11)$$

Like the time-invariant part, the evolution and prediction can be formulated as follows:

$$\begin{aligned} \boldsymbol{z}_{i-1} &= g(\boldsymbol{x}_{i-1}), \quad \boldsymbol{z}_i = \mathcal{K}_{\text{ns}}\boldsymbol{z}_{i-1}, \quad \hat{\boldsymbol{x}}_i = \Psi_{\text{de}}(\boldsymbol{z}_i), i = 2, \ldots, Q, \\ \mathcal{K}_{\text{ns}} &= (\boldsymbol{Z}^{\text{back}})^{\top}(\boldsymbol{Z}^{\text{fore}}), \quad \boldsymbol{Z}^{\text{back}} = [\boldsymbol{z}_1, \boldsymbol{z}_2, \ldots, \boldsymbol{z}_{Q-1}], \quad \boldsymbol{Z}^{\text{fore}} = [\boldsymbol{z}_2, \boldsymbol{z}_3, \ldots, \boldsymbol{z}_Q], \end{aligned} \qquad (12)$$

where $\hat{\boldsymbol{x}}_i$ denotes the forecasting, corresponding the ground trues $\boldsymbol{x}_i$ in eq. (11). $g \in \mathcal{H}$ also acts as a encoder, and $\Psi_{\text{de}}$ denotes the decoder.

### 3.3 DISTRIBUTION CONSTRAINT

A fundamental challenge with non-stationary time series is the time-varying distribution, which refers to the distribution of the time series $P(\boldsymbol{x}_t)$ varying across different time steps $t$. To tackle this challenge, we introduce a constraint module to align the distribution of forecasts with the distribution predicted based on the historical data. Similar to the process in section 3.2, we mine the distribution dynamics in the stationary and non-stationary components respectively. We assume the time series distribution to be Gaussian, as it is omnipresent and enables our method to be tractable.

For the stationary component, we assume that the distribution $\mathcal{N}(\boldsymbol{\mu}_s, \boldsymbol{\delta}_s^2)$ of $\boldsymbol{X}_s$ is constant, where $\boldsymbol{\mu}_s \in \mathbb{R}^{C \times 1}$ is the mean vector, and $\boldsymbol{\delta}_s^2 \in \mathbb{R}^{C \times 1}$ is the variance vector. This assumption means that the forecasting $\boldsymbol{X}_s^{\text{fore}}$ in this component follows the same distribution over time.

For the non-stationary component, we set the distribution sequence of the segmentation in eq. (11) to be $\{\mathcal{N}_{\text{ns}}^1, \mathcal{N}_{\text{ns}}^2, \ldots, \mathcal{N}_{\text{ns}}^Q\}$, and $\boldsymbol{x}_i \sim \mathcal{N}_{\text{ns}}^i = \mathcal{N}(\boldsymbol{\mu}_{\text{ns}}^i, \boldsymbol{\delta}_{\text{ns}}^{2,i})$. Describe the evolution of the distribution by a Koopman operator $\mathcal{K}_{\text{dis}}$, such that:

$$g_{\text{dis}}[\boldsymbol{\mu}_{\text{ns}}^i, \boldsymbol{\delta}_{\text{ns}}^{2,i}] = \mathcal{K}_{\text{dis}}g_{\text{dis}}[\boldsymbol{\mu}_{\text{ns}}^{i-1}, \boldsymbol{\delta}_{\text{ns}}^{2,i-1}], i = 2, 3, \ldots, Q, \qquad (13)$$

where $g_{\text{dis}} \in \bar{\mathcal{H}}$ is the encoder (*i.e.,* the measurement function) for the distributions. There is also a corresponding decoder $\Upsilon_{\text{de}}$ makes the predicted distribution $[\hat{\boldsymbol{\mu}}_{\text{ns}}^i, \hat{\boldsymbol{\delta}}_{\text{ns}}^{2,i}] = \Upsilon_{\text{de}}(\mathcal{K}_{\text{dis}}g_{\text{dis}}[\boldsymbol{\mu}_{\text{ns}}^{i-1}, \boldsymbol{\delta}_{\text{ns}}^{2,i-1}])$.

In particular, Sinkhorn loss (Cuturi, 2013) is utilized to quantify the disparity between the predicted distribution and the ground truth, enabling distribution alignment. It is defined as follows:

$$\mathcal{L}_{\text{dis}} = \mathcal{L}(\mathcal{N}^{\text{gt}}, \mathcal{N}^{\text{fore}}) = \min(L \odot P - \gamma E(P)), \qquad (14)$$

where $\mathcal{N}_{\text{ns}}^{\text{gt}}$ and $\mathcal{N}_{\text{ns}}^{\text{fore}}$ denote the ground truth and the forecasting of the data distribution. $\boldsymbol{L}$ denotes the loss matrix. $\boldsymbol{P}$ denotes a transition matrix. $\odot$ denotes the Hadamard product. $\gamma$ is a regularization parameter. $E(\boldsymbol{P})$ denotes the entropy of the transition matrix $\boldsymbol{P}$.

## 4 EXPERIMENT

In this section, we evaluate the performance of the proposed KokerNet on several commonly used benchmarks and examine the influence of the stationarity of the time series on the decomposition using the introduced index. We first introduce the implementation details, including comparison methods, evaluation datasets, and experiment settings. Then, we systemically conduct experiments, and the results consistently demonstrate the superiority of KokerNet.

### 4.1 DATASETS AND COMPARED METHODS

**Datasets** For the multivariate time series forecasting, we include six real-world time series datasets: **ETT** (Electricity Transformer Temperature) (Zhou et al., 2021), which consists of 2 years data from two separated counties in China and also include different subsets, {ETTh1, ETTh2} for 1-hour-level and {ETTm1, ETTm2} for 15-minutes-level; **Exchang** (Lai et al., 2018), collecting the panel data of daily exchange rates from 8 countries from 1990 to 2016; **ECL** (Electricity Consuming Load)[1], which records the hourly electricity consumption of 321 clients from 2012 to 2014; **ILI** (Influenza-like Illness)[2] collects the ratio of influenza-like illness patients versus the total patients in one week, which is reported weekly by Centers for Disease Control and Prevention of the United States from 2002 and 2021; **Traffic** (PeMS) and **Weather**(Wetterstation) (Liu et al., 2023). For the univariate time series forecasting, **M4** dataset is applied (Makridakis et al., 2018) to evaluate the performance of the proposed KokerNet. It is a collection of 100000 time series used for the fourth edition of the Makridakis forecasting competition and contains time series with different frequencies (hourly, daily, weekly, monthly, quarterly, and yearly).

**Compared methods** For the multivariate time series forecasting, we compare six state-of-the-art forecasting approaches, including **Autoformer** (Wu et al., 2021), **Non-stationary Transformer** (Liu et al., 2022), **Crossformer** (Zhang & Yan, 2022), **iTransformer** (Liu et al., 2024), **KNF** (Wang et al., 2023b), and **Koopa** (Liu et al., 2023). For the univariate time series forecasting, we compare three state-of-the-art approaches, including **PatchTST** (Nie et al., 2022), **DLinear** (Zeng et al., 2023) and **Koopa** (Liu et al., 2023).

### 4.2 IMPLEMENTATION DETAILS

All the experiments are implemented using PyTorch (Paszke et al., 2019) and conducted on a workstation with NVIDIA RTX 3090 GPU, AMD R7-5700X 3.40GHz 8-core CPU, and 32 GB memory. Each method is trained by the ADAM (Kingma & Ba, 2015) algorithm. The loss function consists of three components, the forecasting loss $\mathcal{L}_{\text{fore}}$, the reconstruction loss $\mathcal{L}_{\text{rec}}$, and the distribution loss $\mathcal{L}_{\text{dis}}$. For the forecasting and reconstruction losses $\mathcal{L}_{\text{fore}}$, $\mathcal{L}_{\text{rec}}$, mean square error (MSE) loss is selected to optimize the model parameters, while the Sinkhorn loss (Cuturi, 2013) is applied in the distribution loss $\mathcal{L}_{\text{dis}}$.

In the experiment, we set the lookback length $T = 2H$, meaning the number of segments $Q = 2$. The number of forecasting steps is set to $h = 1$, due to the non-stationary property of the time series. The decoders are the multi-layer perceptions (MLP) with 2 hidden layers, using the tanh activation function. Other hyper-parameters, such as the learning rate and the top percent $\alpha$, in each dataset are different. For multivariate time series forecasting, mean square error (MSE) and mean absolute error (MAE) are used to assess the performance of different methods. For the univariate time series forecasting, symmetric mean absolute percentage error (sMAPE) (Makridakis, 1993), mean absolute percentage error (MAPE), and mean absolute scaled error (MASE) (Hyndman & Koehler, 2006) are used.

### 4.3 RESULTS

**Multivariate Forecasting Result** For multivariate time series forecasting, we compare our proposed model, KokerNet, with several state-of-the-art models on six commonly used benchmarks. The results, presented in Table 1, demonstrate that KokerNet achieves remarkable performance

---

[1]https://archive.ics.uci.edu/ml/datasets/ ElectricityLoadDiagrams20112014
[2]https://gis.cdc.gov/grasp/fluview/fluportaldashboard.html

Table 1: Multivariate time series forecasting results with different forecasting lengths $H \in \{24, 36, 48, 60\}$ for ILI dataset and $H \in \{48, 96, 144, 192\}$ for others under $T = 2H$. The best results are highlighted in **bold** and the suboptimal results are highlighted in underline. Additional results (ETTm1, ETTm2, ETTh1) are provided in appendix E.1. (All results of the compared methods are replications based on the publicly available code.)

| Models | Metric | KokerNet | | Ns_Transformer | | Autoformer | | Koopa | | iTransformer | | KNF | | Crossformer | |
|---|---|---|---|---|---|---|---|---|---|---|---|---|---|---|---|
| | | MSE | MAE | MSE | MAE | MSE | MAE | MSE | MAE | MSE | MAE | MSE | MAE | MSE | MAE |
| ETTh2 | 48 | **0.2299** | **0.3036** | 0.3096 | 0.3724 | 0.3072 | 0.3671 | 0.2434 | 0.3107 | 0.2374 | 0.3105 | 0.3850 | 0.3760 | 0.3557 | 0.4067 |
| | 96 | **0.2929** | **0.3467** | 0.4121 | 0.4332 | 0.3686 | 0.4113 | 0.3046 | 0.3562 | 0.3083 | 0.3593 | 0.4330 | 0.4460 | 0.5568 | 0.5502 |
| | 144 | **0.3277** | **0.3727** | 0.4697 | 0.4533 | 0.4036 | 0.4267 | 0.3404 | 0.3874 | 0.3441 | 0.3849 | 0.4410 | 0.4560 | 0.6442 | 0.5972 |
| | 192 | **0.3575** | **0.3927** | 0.5521 | 0.4964 | 0.4183 | 0.4314 | 0.3543 | 0.3954 | 0.3678 | 0.4015 | 0.5280 | 0.5030 | 1.2161 | 0.8395 |
| Traffic | 48 | **0.4458** | **0.2928** | 0.6067 | 0.3351 | 0.6105 | 0.3851 | 0.4818 | 0.3309 | 0.4998 | 0.3491 | 0.6210 | 0.3820 | 1.3297 | 0.7859 |
| | 96 | **0.4089** | **0.2816** | 0.6165 | 0.3463 | 0.6510 | 0.3978 | 0.5342 | 0.3595 | 0.4506 | 0.3239 | 0.6450 | 0.3760 | 1.3033 | 0.7858 |
| | 144 | **0.4089** | **0.2863** | 0.6206 | 0.3468 | 0.6941 | 0.4198 | 0.5180 | 0.3526 | 0.4526 | 0.3310 | 0.6830 | 0.4020 | 1.3099 | 0.7908 |
| | 192 | **0.4159** | **0.2914** | 0.6336 | 0.3497 | 0.6595 | 0.4143 | 0.5235 | 0.3555 | 0.4601 | 0.3383 | 0.6990 | 0.4050 | 1.3183 | 0.7921 |
| Weather | 48 | 0.1398 | 0.1777 | 0.1413 | 0.1888 | 0.2946 | 0.3521 | 0.1253 | 0.1667 | 0.1367 | 0.1729 | 0.2010 | 0.2880 | 0.1359 | 0.1988 |
| | 96 | 0.1664 | 0.2112 | 0.1907 | 0.2373 | 0.2943 | 0.3606 | 0.1592 | 0.2051 | 0.1694 | 0.2152 | 0.2950 | 0.3080 | 0.1664 | 0.2302 |
| | 144 | **0.1830** | 0.2307 | 0.2244 | 0.2665 | 0.2941 | 0.3522 | 0.1842 | 0.2282 | 0.1880 | 0.2351 | 0.3940 | 0.4010 | 0.1911 | 0.2631 |
| | 192 | 0.2031 | 0.2509 | 0.2350 | 0.2775 | 0.3171 | 0.3749 | 0.2081 | 0.2495 | 0.2033 | 0.2501 | 0.4620 | 0.4370 | 0.2105 | 0.2759 |
| Exchange | 48 | 0.0452 | 0.1469 | 0.0645 | 0.1780 | 0.1117 | 0.2458 | 0.0415 | 0.1518 | 0.0458 | 0.1502 | 0.1280 | 0.2710 | 0.1823 | 0.2993 |
| | 96 | 0.0897 | 0.2118 | 0.1552 | 0.2705 | 0.1498 | 0.2793 | 0.0916 | 0.2118 | 0.0974 | 0.2225 | 0.2940 | 0.3940 | 0.3026 | 0.4018 |
| | 144 | 0.1398 | 0.2697 | 0.1859 | 0.3103 | 0.2096 | 0.3339 | 0.1351 | 0.2607 | 0.1512 | 0.2788 | 0.5970 | 0.5780 | 0.4056 | 0.4809 |
| | 192 | **0.1862** | **0.3090** | 0.2489 | 0.3642 | 0.2794 | 0.3843 | 0.1892 | 0.3136 | 0.2075 | 0.3303 | 0.6540 | 0.5950 | 0.5464 | 0.5889 |
| ILI | 24 | **1.8710** | **0.8351** | 2.2136 | 0.9266 | 3.9796 | 1.3951 | 2.0618 | 0.8790 | 4.0837 | 1.4320 | 3.7220 | 1.4320 | 3.7474 | 1.3762 |
| | 36 | 1.9181 | 0.8934 | 2.5677 | 0.9452 | 3.5755 | 1.2982 | 1.8611 | 0.8946 | 4.3227 | 1.4926 | 3.9410 | 1.4480 | 4.0997 | 1.3809 |
| | 48 | 1.8849 | 0.9223 | 2.6007 | 1.0192 | 3.2697 | 1.2443 | 1.9033 | 0.9239 | 4.0271 | 1.4459 | 3.2870 | 1.3770 | 3.7599 | 1.2884 |
| | 60 | 1.9347 | 0.9583 | 2.5717 | 1.0255 | 3.4445 | 1.2746 | 1.8502 | 0.8843 | 4.4316 | 1.5186 | 2.9740 | 1.3010 | 4.4182 | 1.4273 |
| ECL | 48 | 0.1570 | 0.2437 | 0.1535 | 0.2597 | 0.1893 | 0.3061 | 0.1280 | 0.2302 | 0.1512 | 0.2423 | 0.1750 | 0.2650 | 0.1484 | 0.2515 |
| | 96 | 0.1365 | 0.2291 | 0.1785 | 0.2843 | 0.2042 | 0.3206 | 0.1389 | 0.2387 | 0.1375 | 0.2322 | 0.1980 | 0.2840 | 0.1335 | 0.2282 |
| | 144 | 0.1453 | 0.2373 | 0.1889 | 0.2919 | 0.2021 | 0.3153 | 0.1518 | 0.2515 | 0.1576 | 0.2424 | 0.2040 | 0.2970 | 0.1557 | 0.2546 |
| | 192 | **0.1520** | **0.2441** | 0.1967 | 0.3025 | 0.2133 | 0.3288 | 0.1566 | 0.2556 | 0.1545 | 0.2503 | 0.2450 | 0.3210 | 0.1557 | 0.2507 |

Table 2: Univariate time series forecasting results with different frequencies on **M4** dataset. The best results are highlighted in **bold**. (All the results of the compared methods are replications based on the publicly available code.)

| | KokerNet | | | PatchTST | | | DLinear | | | Koopa | | |
|---|---|---|---|---|---|---|---|---|---|---|---|---|
| | sMAPE | MAPE | MASE | sMAPE | MAPE | MASE | sMAPE | MAPE | MASE | sMAPE | MAPE | MASE |
| Yearly | **13.454** | **16.571** | **3.033** | 16.668 | 23.302 | 3.729 | 15.413 | 18.467 | 3.696 | 14.707 | 19.417 | 3.275 |
| Quarterly | **10.213** | **11.779** | **1.192** | 12.606 | 15.118 | 1.628 | 10.546 | 12.288 | 1.242 | 10.775 | 12.823 | 1.287 |
| Monthly | **12.780** | **14.874** | **0.940** | 15.859 | 19.902 | 1.273 | 13.233 | 15.750 | 0.985 | 16.127 | 19.378 | 1.270 |
| Weekly | 11.157 | 10.309 | 3.396 | 11.551 | 11.234 | 4.465 | 11.168 | 12.003 | 5.936 | **10.221** | **9.542** | **3.135** |
| Daily | **3.035** | 4.387 | 3.251 | 3.576 | 5.590 | 3.894 | 3.384 | 5.165 | 3.685 | 3.395 | 4.886 | 3.682 |
| Hourly | 18.013 | 23.685 | 3.094 | 34.211 | 118.404 | 10.752 | **17.223** | **23.482** | **2.702** | 18.171 | 23.683 | 2.808 |
| Others | **4.858** | **6.410** | **3.248** | 6.685 | 15.336 | 4.503 | 5.089 | 7.173 | 3.765 | 5.109 | 6.777 | 3.570 |
| Average | **12.408** | **14.739** | **1.623** | 15.474 | 20.841 | 2.535 | 13.269 | 16.089 | 2.196 | 14.476 | 17.862 | 2.207 |

on most datasets. Specifically, KokerNet consistently outperforms the state-of-the-art transformer-based non-stationary models (*i.e., Ns_Transformer*), highlighting the Koopman-based model is more adept at exploring the non-stationarity of the time series. Compared to Koopman-based counterparts (Koopa and KNF), Kokernet achieves superior performance. This success derives from the distribution constraint, enabling the exploration of time-varying distributions in non-stationary time series. Note that Koopa is comparable to our method in short-term time series forecasting. This is because Koopa also deems that the time series consists of both time-invariant and time-variant components and then designs the global shared and local Koopman operators, respectively. However, for long-term forecasting, KokerNet is superior to Koopa, which is attributed to the constructed measurement function space and the distribution constraint. On one hand, the measurement functions are higher-order derivatives, which enables us to supervise any derivative of the measurement function with complicated time series. On the other hand, the distribution constraint ensures that the prediction aligns with the temporal variation in distribution, enabling the exploration of long-term non-stationary time series.

**Univariate Forecasting Results** For univariate time series forecasting, we compare our Koker-Net with three state-of-the-art models on the **M4** dataset. The results, displayed in Table 2, illustrate the general superiority of our model. Take the sMAPE metric for example, we can observe that KokerNet exhibits enhanced performance in scenarios where seasonality is less pronounced and forecastability is heightened (*e.g.,* our KokerNet achieves a 12.71% reduction in sMAPE for *Yearly* data and a 10.31% reduction for *Daily* data). By contrast, KokerNet tends to perform mediocre in scenarios where seasonality is more pronounced, and achieves 3.60% sMAPE reduction for *Quarterly* data, 3.42% reduction for *Monthly* data. For instance of perfect seasonality, the performance of the proposed KokerNet is even worse than the baseline models (achieving 4.53% increase in

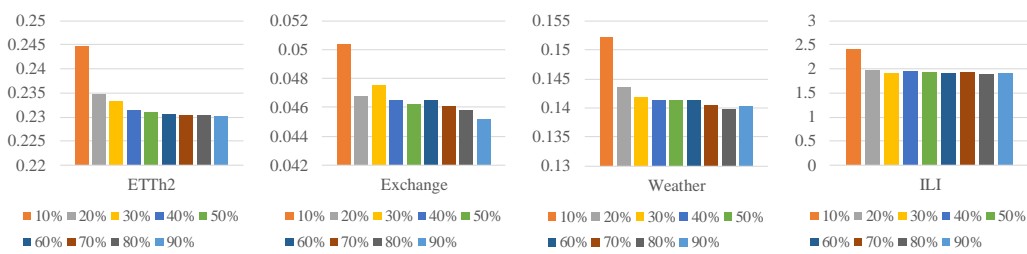

Figure 1: The influence of the proportion of the stationary components on the results.

Table 3: The results with the single component. Here, $\mathcal{K}_s$ denotes only the global shared Koopman operator included in our model, $\mathcal{K}_{ns}$ denotes only local Koopman operator included in our model, and $\text{Pre}_{def}$ denotes the best result for different proportion of the stationary component under the decomposition case. The best results are highlighted in **bold**.

| | | $\mathcal{K}_s$ | | | | $\mathcal{K}_{ns}$ | | | | $\text{Pre}_{def}$ | | | |
|---|---|---|---|---|---|---|---|---|---|---|---|---|---|
| | | 48(24) | 96(36) | 144(48) | 192(60) | 48(24) | 96(36) | 144(48) | 192(60) | 48(24) | 96(36) | 144(48) | 192(60) |
| ETTh2 | MSE | 0.2299 | **0.2929** | **0.3277** | **0.3655** | 0.3064 | 0.3665 | 0.3910 | 0.3914 | 0.2303 | 0.2960 | 0.3336 | 0.3579 |
| | MAE | **0.3036** | **0.3467** | **0.3727** | **0.3986** | 0.3629 | 0.3999 | 0.4232 | 0.4256 | 0.3043 | 0.3499 | 0.3766 | 0.3929 |
| ILI | MSE | 1.8983 | **1.9167** | 1.9811 | 1.9916 | 4.1361 | 3.8204 | 4.0049 | 4.2990 | **1.8710** | 1.9181 | **1.8849** | **1.9347** |
| | MAE | 0.8611 | 0.8950 | 0.9559 | 0.9763 | 1.5025 | 1.4293 | 1.4970 | 1.5263 | **0.8351** | **0.8934** | **0.9223** | **0.9583** |

sMAPE). This phenomenon may be caused by the distribution constraint, which is less relevant for more stationary data.

Besides, to evaluate the model efficiency, we take three aspects into account, including forecasting performance (MSE), training time, and memory footprint. We compare the models on the **ETTh1** dataset with the forecasting length $H = 144$. The results are reported in appendix E.4, showing that our KokerNet has better forecasting performance with less training time and memory footprint.

## 4.4 TIME SERIES DECOMPOSITION

As previously discussed, the real-world time series typically contains both time-invariant and time-variant patterns, corresponding to the stationary and non-stationary components. Therefore, we decompose the time series into these two components to evade information loss and the introduction of unnecessary disturbances from the single-component assumption. To interpretably determine the proportions of each component, an index $S_v$ is designed based on the KS test.

To validate the guiding role of $S_v$ in time series decomposition, we first calculate the value of $S_v$ on four datasets (ETTh2, Traffic, Weather, and ILI), with results $S_v^{\text{ETTh2}} = 0.0865$, $S_v^{\text{Traffic}} = 0.0767$, $S_v^{\text{Weather}} = 0.3709$, and $S_v^{\text{ILI}} = 0.2653$, where the length of each segmentation equals $H = 48$. Then, we set the candidate range of the proportions as $[10\%, 20\%, \ldots, 90\%]$, and report MSE under different proportions, where $10\%$ represents that the stationary component account for $10\%$, while the non-stationary component account for $90\%$. Results are shown in Figure 1. We can observe that as the proportion of stationary components increases, the MSE tends to decrease, which is attributed to the scarcity of non-stationary components. The smaller the value of $S_v$, the more pronounced this phenomenon. For example, $S_v^{\text{ETTh2}} = 0.0865$, indicating that the proportion of non-stationary components is minimal. In this scenario, increasing the proportion of non-stationary components would introduce uncertainty to the stationary component, resulting in unreliable results. In contrast, $S_v^{\text{Weather}} = 0.3709$, meaning this dataset includes more non-stationary components. We can observe that the result at the $80\%$ is the most optimal, and it will decrease when increasing the proportion of the stationary component. That is because increasing the proportion of the stationary component would lead to the over-stationary for the non-stationary component, resulting in suboptimal performance. Thus, we can suggest that the designed index $S_v$ effectively measures the stationarity of the time series and guides the decomposition.

Furthermore, we evaluate the effectiveness of $S_v$ via considering three cases, including $\mathcal{K}_s$, $\mathcal{K}_{ns}$, and $\text{Pre}_{def}$. $\mathcal{K}_s$ denotes the entire time series is stationary, and only the global Koopman operator is learned for the time series. $\mathcal{K}_{ns}$ denotes the entire time series is non-stationary, and only the local

Table 4: Ablation study for distribution constraint. The best results are highlighted in **bold**.

| | Yearly | | | Quarterly | | | Monthly | | | Daily | | |
| | sMAPE | MAPE | MASE | sMAPE | MAPE | MASE | sMAPE | MAPE | MASE | sMAPE | MAPE | MASE |
|---|---|---|---|---|---|---|---|---|---|---|---|---|
| w/ $\mathcal{K}_{dis}$ | **13.454** | **16.571** | **3.033** | **10.213** | **11.799** | **1.192** | **12.780** | **14.874** | **0.940** | **3.035** | **4.387** | **3.251** |
| w/o $\mathcal{K}_{dis}$ | 14.277 | 17.099 | 3.096 | 10.408 | 12.051 | 1.219 | 13.066 | 15.399 | 0.974 | 3.080 | 4.408 | 3.319 |
| Promotion | 5.76% | 3.09% | 2.03% | 1.87% | 2.09% | 2.21% | 2.19% | 3.41% | 3.49% | 1.46% | 0.68% | 2.05% |

Koopman operator is learned. Specifically, in the $\text{Pre}_{\text{def}}$ case, we set the candidate range of the proportions as $[10\%, 20\%, \ldots, 90\%]$ and select the most optimal result, where $10\%$ represents that the stationary component account for $10\%$, while the non-stationary component account for $90\%$. The results under different cases are reported in Table 3.

The results in Table 3 show that: 1) For the time series with few non-stationary components, it is not necessary to perform the decomposition. For **ETTh2**, the best result is the $\mathcal{K}_s$ case since this data contains more stationary components with $S_v^{\text{ETTh2}} = 0.0865$; 2) For the time series with more non-stationary components (such as **ILI** with $S_v^{\text{ILI}} = 0.2653$), the decomposition will improve the performance of the model, which is ascribed to specialized Koopman operators that are learned for different components. In addition, we can observe that the performance of $\text{Pre}_{\text{def}}$ is commonly close to our KokerNet. It is because $\text{Pre}_{\text{def}}$ select the best result of all the candidate proportions. But for the time series with few non-stationary components, KokerNet performs better than $\text{Pre}_{\text{def}}$ under the guiding of index $S_v$. That result further demonstrates the reliability of the designed index $S_v$ and the importance of performing time series decomposition.

### 4.5 ABLATION STUDY

A fundamental challenge with deep forecasting models is that the non-stationary information extracted from historical data does not consistently align with predictions. Therefore, we introduce a distribution constraint for the non-stationary component to align the forecasting with the evolution in distribution. To demonstrate the effectiveness of the distribution constraint, we conduct an ablation study on **M4** under two cases, with (w/) and without (w/o) the distribution constraint. The results, reported in Table 4, show that the case w/ $\mathcal{K}_{\text{dis}}$ consistently performs better on different frequencies. Specifically, for scenarios with weak seasonality, such as with the frequency *Yearly*, the performance improvement brought by the incorporation of distribution constraint is more pronounced. This further highlights the effectiveness of the distribution constraint.

## 5 CONCLUSION

In this paper, we propose a novel method, KokerNet, for time series forecasting. In the method, a measurement function space is first constructed based on the spectral kernel methods to learn the Koopman operator, which describes the dynamics of the time series. Then, an index is designed to guide the time series decomposition, and global and local operators are further learned based on the decomposition within the constructed measurement space. Finally, our model incorporates a distribution constraint module to ensure the prediction aligns with the temporal variation in distribution. Theoretical analysis and extensive experiments demonstrate that the proposed approach delivers significant performance improvements. We believe our approach can offer a new perspective on time series forecasting.

**Limitation and Future Work** KokerNet tends to learn global and local Koopman operators for the stationary and non-stationary components, which are obtained by the decomposition based on the value of $S_v$. However, we can not calculate the stationarity of a real-world time series accurately. In future work, we will focus on measuring the stationarity of the real-world time series more precisely. In this paper, we model the complex evolution of the time series as its temporal dependence using the kernel method, with the measurement function $g \in \mathcal{H}$ serving as an encoder that maps the time series into a low-dimensional feature space. Deep kernel, sharing the advantages of deep learning and kernel method, can be considered as the encoder to capture the complicated temporal dependence in the future. In addition, we assume that the distribution of the time series is Gaussian to make our method tractable. In the future, the joint distribution can be considered to take both the interactions between variables and the variation of the distribution via the Copula function.

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

# APPENDIX

## A  ALGORITHM

---

**Algorithm 1** KokerNet for time series forecasting.

---

**Input:** $\boldsymbol{X}_T$ with $T$ time points.

**Output:** $\mathcal{K}_\text{s}, \mathcal{K}_\text{ns}, \mathcal{K}_\text{dis}, g_{\boldsymbol{\Theta}}(\cdot), \boldsymbol{\Theta} = \{\omega_1, \cdots, \omega_M\}, \Phi_\text{de}, \Psi_\text{de}.$

1: Calculating $S_v \leftarrow p\bar{p}$ based on Eq (7) and Eq (9).

2: Dividing $\boldsymbol{X}_T$ into $\boldsymbol{X}_\text{s}$ and $\boldsymbol{X}_\text{ns}$ based on $S_v$, and

3: **repeat**

4:     **For** $\boldsymbol{X}_\text{s} = [\boldsymbol{X}_\text{s}^\textbf{back}, \boldsymbol{X}_\text{s}^\textbf{fore}]$:

5:     Compute the distribution $\mathcal{N}(\boldsymbol{\mu}_\text{s}, \boldsymbol{\delta}_\text{s}^2) \leftarrow \boldsymbol{X}_\text{s}^\text{back}$;

6:     Compute $\boldsymbol{Z}_\text{s}^\text{back} \leftarrow g(\boldsymbol{X}_\text{s}^\text{back})$, and forecast $\boldsymbol{Z}_\text{s}^\text{fore} \leftarrow \mathcal{K}_\text{s}\boldsymbol{Z}_\text{s}^\text{back}$ with $\mathcal{K}_\text{s}$;

7:     Decode $\boldsymbol{Z}_\text{s}^\text{fore}$ with the decoder $\Phi_\text{de}$, $\hat{\boldsymbol{X}}_\text{s}^\text{fore} \leftarrow \Phi_\text{de}(\boldsymbol{Z}_\text{s}^\text{fore})$;

8:     Compute the distribution $\mathcal{N}(\hat{\boldsymbol{\mu}}_\text{s}, \hat{\boldsymbol{\delta}}_\text{s}^2) \leftarrow \hat{\boldsymbol{X}}_\text{s}^\text{fore}$;

9:     Compute the loss $\mathcal{L}_\text{fore}^\text{s}; \mathcal{L}_\text{dis}^\text{s}$.

10:

11:     **For** $\boldsymbol{X}_\text{ns} = [\boldsymbol{x}_1, \cdots, \boldsymbol{x}_Q]$:

12:     Compute the distribution $\{\mathcal{N}_\text{ns}^1, \mathcal{N}_\text{ns}^2, \ldots, \mathcal{N}_\text{ns}^Q\} \leftarrow \boldsymbol{X}_\text{ns} = [\boldsymbol{x}_1, \cdots, \boldsymbol{x}_Q]$;

13:     $\boldsymbol{z}_i \leftarrow g(\boldsymbol{x}_i), i = 1, \ldots, Q$;

14:     $\boldsymbol{Z}^\text{back} \leftarrow [\boldsymbol{z}_1, \ldots, \boldsymbol{z}_{Q-1}], \boldsymbol{Z}^\text{fore} \leftarrow [\boldsymbol{z}_2, \ldots, \boldsymbol{z}_Q]$;

15:     $\mathcal{K}_\text{ns} \leftarrow (\boldsymbol{Z}^\text{back})(\boldsymbol{Z}^\text{fore})^\top$;

16:     $\hat{\boldsymbol{z}}_i \leftarrow \mathcal{K}_\text{ns}\boldsymbol{z}_{i-1}$

17:     Decode $\hat{\boldsymbol{x}}_i \leftarrow \Psi(\hat{\boldsymbol{z}}_i), i = 2, \ldots, Q$;

18:     Compute the distribution $\mathcal{N}(\hat{\boldsymbol{\mu}}_i, \hat{\boldsymbol{\delta}}_i^2) \leftarrow \hat{\boldsymbol{x}}_i, i = 2, \ldots, Q$;

19:     Compute the loss $\mathcal{L}_\text{fore}^\text{ns}; \mathcal{L}_\text{dis}^\text{alig} \leftarrow \mathcal{L}(\mathcal{N}(\hat{\boldsymbol{\mu}}_i, \hat{\boldsymbol{\delta}}_i^2), \mathcal{N}(\hat{\boldsymbol{\mu}}_\text{ns}^i, \hat{\boldsymbol{\delta}}_\text{ns}^{2,i})), i = 2, \ldots, Q$.

20:

21:     **While for the distribution do**:

22:     $g_\text{dis}[\boldsymbol{\mu}_\text{ns}^i, \boldsymbol{\delta}_\text{ns}^{2,i}] \leftarrow \mathcal{N}(\boldsymbol{\mu}_\text{ns}^i, \boldsymbol{\delta}_\text{ns}^{2,i}), i = 1, \cdots, Q$;

23:     Similar to the process of $\boldsymbol{X}_\text{ns}$;

24:     Decode $[\hat{\boldsymbol{\mu}}_\text{ns}^i, \hat{\boldsymbol{\delta}}_\text{ns}^{2,i}] \leftarrow \Upsilon_\text{de}(\mathcal{K}_\text{dis}[\boldsymbol{\mu}_\text{ns}^{i-1}, \boldsymbol{\delta}_\text{ns}^{2,i-1}]), i = 2, \cdots, Q$;

25:     Compute the loss $\mathcal{L}_\text{dis}^\text{ns} \leftarrow \mathcal{L}(\mathcal{N}(\boldsymbol{\mu}_\text{ns}^i, \boldsymbol{\delta}_\text{ns}^{2,i}), \mathcal{N}(\hat{\boldsymbol{\mu}}_\text{ns}^i, \hat{\boldsymbol{\delta}}_\text{ns}^{2,i})), i = 2, \cdots, Q$.

26:

27:     Compute the total loss $\mathcal{L}_\text{KokerNet} \leftarrow \mathcal{L}_\text{fore}^\text{s} + \mathcal{L}_\text{dis}^\text{s} + \mathcal{L}_\text{fore}^\text{ns} + \mathcal{L}_\text{dis}^\text{ns} + \mathcal{L}_\text{dis}^\text{alig} + \mathcal{L}_\text{rec}$.

28:     **Update**

29: **until** Convergence

---

## B  RELATED WORKS IN NON-STATIONARY TIME SERIES FORECASTING

Transformer-based deep models (Zhou et al., 2021; Wu et al., 2021; Zhang & Yan, 2022; Zhou et al., 2022; Liu et al., 2021) have achieved great success in forecasting time series with seasonality and trend. However, most of these models are difficult to deal with the non-stationary time series, characterized by the intrinsic change of distribution over time. Recently, several approaches to non-stationary time series forecasting have been developed (Passalis et al., 2019; Kim et al., 2021; Liu et al., 2022). These approaches can be roughly categorized into two aspects. One is the stationar-ization method, where the focus is on processing the non-stationary time series into stationary ones before performing the forecasting task. Adaptive Norm (Ogasawara et al., 2010) applies z-score normalization for each series fragment by global statistics of a sampled set. DAIN (Passalis et al., 2019) employs a nonlinear neural network to adaptively stationarize time series according to the observed training distribution. RevIN (Kim et al., 2021) introduces a two-stage instance normaliza-tion, which transforms model input and output respectively to reduce the discrepancy of each series. Non-stationary Transformer (Liu et al., 2022) utilizes series stationarization to attenuate time series non-stationarity and de-stationary attention to re-incorporate non-stationary information of raw se-ries. The other category is decomposition methods, which divides the non-stationary time series into

time-invariant and time-variant parts (Wang et al., 2023; Liu et al., 2023). The time-invariant part is used to characterize the shared global dynamics, while the time-variant part is used to describe the localized dynamics. KNF (Wang et al., 2023) and Koopa (Liu et al., 2023) cope with the non-stationary time series by introducing both the global and local Koopman operators to explore the time-invariant and time-variant dynamics, respectively.

## C  THE ARCHITECTURE OF KOOPMAN OPERATOR LEARNING AND DISTRIBUTION CONSTRAINT

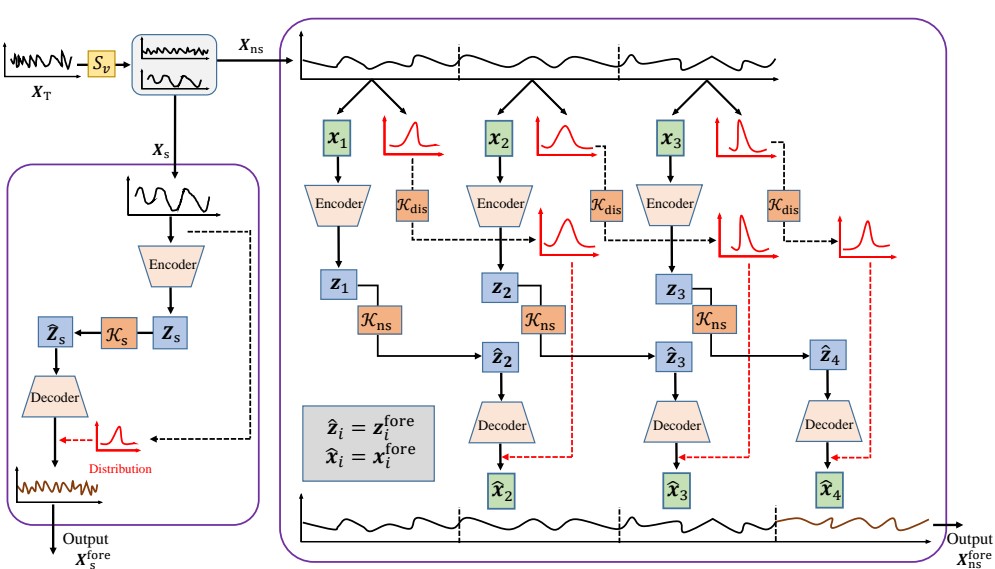

Figure 4: The architecture of Koopman operator learning and distribution constraint. We first decompose the time series $X_T$ into stationary and non-stationary components based on the designed index $S_v$. For the stationary input $X_s$, a global Koopman operator $\mathcal{K}_s$ is learned with the stationary distribution constraint. For the non-stationary input $X_{ns}$, a local Koopman operator $\mathcal{K}_{ns}$ is learned with the non-stationary distribution constraint, which is explored by the historical distribution via the distribution Koopman operator $\mathcal{K}_{dis}$.

## D  THE PROOF OF THEORETICAL RESULTS

### D.1  THE DERIVATION PROCEDURE OF $g$ IN EQ. (3)

**Lemma D.1.** *(Bochner's Theorem) A continuous kernel $k(\boldsymbol{x}, \boldsymbol{x}') = k(\boldsymbol{x} - \boldsymbol{x}')$ on $\mathbb{R}^d$ is positive definite if and only if $k(\boldsymbol{\tau}), \boldsymbol{\tau} = \boldsymbol{x} - \boldsymbol{x}'$ is the Fourier transform of a non-negative measure. Such that:*

$$k(\boldsymbol{\tau}) = \int_{\mathbb{R}^d} s(\lambda) e^{i\lambda\boldsymbol{\tau}} d\lambda,$$
$$s(\lambda) = \int_{\mathbb{R}^d} k(\boldsymbol{\tau}) e^{-i\lambda\boldsymbol{\tau}} d\boldsymbol{\tau}. \tag{C.1}$$

Bochner's Theorem ensures that its inverse Fourier Transform is a probability measure, which means that $s(w)$ can be considered as a probability density function.

$$k(\boldsymbol{x} - \boldsymbol{x}') = \int_{\mathbb{R}^d} s(w) e^{i\boldsymbol{w}\boldsymbol{\tau}} d\boldsymbol{w}$$

$$= \mathbb{E}_{\boldsymbol{w} \sim \mathbb{S}}[e^{i\boldsymbol{w}\boldsymbol{\tau}}]$$

$$= \mathbb{E}_{\boldsymbol{w} \sim \mathbb{S}}[\cos \boldsymbol{w}(\boldsymbol{x} - \boldsymbol{x}') + i \sin \boldsymbol{w}(\boldsymbol{x} - \boldsymbol{x}')]$$

$$= \mathbb{E}_{\boldsymbol{w} \sim \mathbb{S}}[\cos \boldsymbol{w}(\boldsymbol{x} - \boldsymbol{x}')] \qquad \text{(C.2)}$$

$$= 2\mathbb{E}_{\varphi \sim [-\pi, \pi]}[\cos (\boldsymbol{w}\boldsymbol{x} + \varphi) \cos (\boldsymbol{w}\boldsymbol{x}' + \varphi)]$$

$$\approx \frac{2}{M} \sum_{m=1}^{M} \langle g(\boldsymbol{x}), g(\boldsymbol{x}') \rangle,$$

where $g(\boldsymbol{x}) = \sqrt{\frac{2}{M}}[\cos(\boldsymbol{w}_1\boldsymbol{x} + b_1), \cos(\boldsymbol{w}_2\boldsymbol{x} + b_2), \ldots, \cos(\boldsymbol{w}_M\boldsymbol{x} + b_M)]^\top$, $M$ is the sampling number.

### D.2 THE PROOF OF THEOREM 1

**Definition D.1.** *We say that a matrix $\boldsymbol{A}$ is a $\Delta$-spectral approximation of another matrix $\boldsymbol{B}$, if $(1 - \Delta)\boldsymbol{B} \preceq \boldsymbol{A} \preceq (1 + \Delta)\boldsymbol{B}$.*

**Lemma D.2.** *Let $\boldsymbol{B}$ be a fixed $d_1 \times d_2$ matrix. Construct a $d_1 \times d_2$ random matrix $\boldsymbol{A}$ that satifies*

$$\mathrm{E}[\boldsymbol{A}] = \boldsymbol{B} \quad and \quad ||\boldsymbol{A}||_2 \leq s. \qquad \text{(C.3)}$$

*Let $\boldsymbol{V}_1$ and $\boldsymbol{V}_2$ be semidefinite upper bounds for the expected squares:*

$$\mathrm{E}[\boldsymbol{A}\boldsymbol{A}^*] \preceq \boldsymbol{V}_1 \quad and \quad \mathrm{E}[\boldsymbol{A}^*\boldsymbol{A}] \preceq \boldsymbol{V}_2. \qquad \text{(C.4)}$$

*Define the quantities*

$$v = max(||\boldsymbol{V}_1||_2, ||\boldsymbol{V}_2||_2) \quad and \quad r = (tr(\boldsymbol{V}_1) + tr(\boldsymbol{V}_2))/v. \qquad \text{(C.5)}$$

*Form the matrix sampling estimator*

$$\bar{\boldsymbol{A}}_n = \frac{1}{n} \sum_{k=1}^{n} \boldsymbol{A}_k, \qquad \text{(C.6)}$$

*where each $\boldsymbol{A}_k$ is an independent copy of $\boldsymbol{A}$. Then, for all $t \geq \sqrt{\frac{v}{n}} + \frac{2s}{3n}$,*

$$Pr(||\bar{\boldsymbol{A}}_n) - \boldsymbol{B}||_2 \geq t) \leq 4rexp(\frac{-nt^2/2}{v + 2st/3}). \qquad \text{(C.7)}$$

**Theorem D.1.** *Sample $\boldsymbol{w}$ according to the spectral density function $p(\boldsymbol{w})$ and set $\boldsymbol{Z} = g(\boldsymbol{X}_T)$. When the sampling number $M \geq \frac{2\delta(3\sqrt{n} + 2\Delta)}{3\Delta^2} ln\frac{8\sqrt{n}}{\rho}$, with the probability of at least $1 - \rho$, $\boldsymbol{Z}\boldsymbol{Z}^\top$ is the $\Delta$-spectral approximation of $\boldsymbol{K} = \langle f(\boldsymbol{X}_T), f(\boldsymbol{X}_T) \rangle_{\mathcal{H}}$.*

*Proof.* Since $k(\cdot, \cdot)$ is a positive definite (PD) kernel function, the corresponding kernel matrix $\boldsymbol{K}$ is a PD matrix. So, the kernel matrix $\boldsymbol{K}$ has its inverse form $\boldsymbol{K}^{-1}$, and it can be conducted the eigendecomposition as $\boldsymbol{K} = Q^\top \Lambda Q = Q^\top \Sigma^2 Q$. $\Sigma$ is a diagonal matrix and the elements are the square root of the eigenvalues of the kernel matrix $\boldsymbol{K}$.

Let $\boldsymbol{K} = Q^\top \Sigma^2 Q$ be an eigendecomposition of $\boldsymbol{K}$, the $\Delta$-spectral approximation can be written as:

$$(1 - \Delta)\boldsymbol{K} \preceq \boldsymbol{Z}\boldsymbol{Z}^\top \preceq (1 + \Delta)\boldsymbol{K}. \qquad \text{(C.8)}$$

Simplifying eq. (C.8) and multiplying by $\Sigma^{-1}Q$ on the left and $Q^\top \Sigma^{-1}$ on the right, we have

$$-\Delta\Sigma^{-1}QQ^\top \Sigma^2 QQ^\top \Sigma^{-1} \preceq \Sigma^{-1}Q\boldsymbol{Z}\boldsymbol{Z}^\top Q^\top \Sigma^{-1} - \Sigma^{-1}Q\boldsymbol{K}Q^\top \Sigma^{-1} \preceq \Delta\Sigma^{-1}QQ^\top \Sigma^2 QQ^\top \Sigma^{-1}, \qquad \text{(C.9)}$$

and it suffices to show that:

$$||\Sigma^{-1}Q\boldsymbol{Z}\boldsymbol{Z}^\top Q^\top \Sigma^{-1} - \Sigma^{-1}Q\boldsymbol{K}Q^\top \Sigma^{-1}||_2 \leq \Delta, \qquad \text{(C.10)}$$

holds with a probability of at least $1 - \rho$.

Let

$$\boldsymbol{Y}_m = \Sigma^{-1} Q \boldsymbol{Z}_m \boldsymbol{Z}_m^\top Q^\top \Sigma^{-1}, \tag{C.11}$$

we have

$$\mathrm{E}[\boldsymbol{Y}_m] = \Sigma^{-1} Q \boldsymbol{Z} \boldsymbol{K} \Sigma^{-1}, \frac{1}{M} \sum_{m=1}^M = \Sigma^{-1} Q \boldsymbol{Z} \boldsymbol{Z}^\top Q^\top \Sigma^{-1}. \tag{C.12}$$

Next, we bound the norm of $\boldsymbol{Y}_m$ and the stable rank $\mathrm{E}[\boldsymbol{Y}_m^2]$. Since $\boldsymbol{Y}_m$ is always a rank one matrix we have

$$\begin{aligned}
||\boldsymbol{Y}_m||_2 =& ||\Sigma^{-1} Q \boldsymbol{Z}_m \boldsymbol{Z}_m^\top Q^\top \Sigma^{-1}||_2 \\
=& tr(\Sigma^{-1} Q \boldsymbol{Z}_m \boldsymbol{Z}_m^\top Q^\top \Sigma^{-1}) \\
=& tr(\boldsymbol{Z}_m^\top Q^\top \Sigma^{-1} \Sigma^{-1} Q \boldsymbol{Z}_m) \\
=& \boldsymbol{Z}_m^\top Q^\top \Sigma^{-2} Q \boldsymbol{Z}_m \\
=& \boldsymbol{Z}_m^\top \boldsymbol{K}^{-1} \boldsymbol{Z}_m = \delta,
\end{aligned} \tag{C.13}$$

and

$$\begin{aligned}
\boldsymbol{Y}_m^2 =& \Sigma^{-1} Q \boldsymbol{Z}_m \boldsymbol{Z}_m^\top Q^\top \Sigma^{-1} \Sigma^{-1} Q \boldsymbol{Z}_m \boldsymbol{Z}_m^\top Q^\top \Sigma^{-1} \\
=& \Sigma^{-1} Q \boldsymbol{Z}_m \boldsymbol{Z}_m^\top \boldsymbol{K}^{-1} \boldsymbol{Z}_m \boldsymbol{Z}_m^\top Q^\top \Sigma^{-1} \\
=& \delta \Sigma^{-1} Q \boldsymbol{Z}_m \boldsymbol{Z}_m^\top Q^\top \Sigma^{-1} \\
=& \delta \boldsymbol{Y}_m.
\end{aligned} \tag{C.14}$$

We calculate $\mathrm{E}[\boldsymbol{Y}_m^2]$ as:

$$\begin{aligned}
\mathrm{E}[\boldsymbol{Y}_m^2] =& \mathrm{E}[\delta \boldsymbol{Y}_m] \\
=& \delta \Sigma^{-1} Q \boldsymbol{K} Q^\top \Sigma^{-1} \\
=& \delta \boldsymbol{I}_n.
\end{aligned} \tag{C.15}$$

According to Lemma D.2, we have

$$Pr(||\frac{1}{M} \sum_{m=1}^M \boldsymbol{Y}_m - \Sigma^{-1} Q \boldsymbol{K} Q^\top \Sigma^{-1}||_2 \geq \Delta) \leq 8\sqrt{n} \exp(\frac{M\Delta^2/2}{\delta\sqrt{n} + 2\delta\Delta/3}). \tag{C.16}$$

Therefore, $\boldsymbol{Z}\boldsymbol{Z}^\top$ is the $\Delta$-spectral approximation of $\boldsymbol{K}$ with the probability of at least $1 - \rho$ with the sampling number $M \geq \frac{2\delta(3\sqrt{n}+2\Delta)}{3\Delta^2} \ln \frac{8\sqrt{n}}{\rho}$.

$\square$

## D.3 THE PROOF OF THEOREM 2

For the Koopman operator $\mathcal{K}^t$, an eigenfunction $f \in L^2(\mu)$ corresponding to that eigenvalue satisfies:

$$\mathcal{K}^t f = e^{i\omega t} f. \tag{C.17}$$

where $\omega$ is a real eigenfrequency.

**Theorem D.2.** *For every eigenfrequency $\omega \in R$ of the Koopman operator $\bar{\mathcal{K}}$, Let $g$ be the measurement function on the finite trajectory $\{\boldsymbol{x}_1, \boldsymbol{x}_2, \ldots, \boldsymbol{x}_T\}$. Then,*

*(i) When $T \geq \sqrt{\frac{2}{M}} \frac{3\omega_{max}}{\epsilon}$, $g_\omega$ can approximate any Koopman eigenfunction with $\epsilon$ accuracy, for $\epsilon > 0$.*

*(ii) $\lim_{T \to \infty} g_\omega$ is an eigenfunction of the Koopman operator $\mathcal{K}$.*

*Proof.* We define

$$g_\omega(\boldsymbol{x}) = \frac{1}{T} \int_{\tau=0}^T g(\boldsymbol{F}^\tau(\boldsymbol{x})) e^{-i\omega\tau} d\tau, \tag{C.18}$$

based on $g \in \bar{\mathcal{H}}$. Let the Koopman operator $\mathcal{K}^t$ acts on $g_\omega(\boldsymbol{x})$, such that

$$
\begin{aligned}
\mathcal{K}^t g_\omega(\boldsymbol{x}) =& \frac{1}{T} \int_{\tau=0}^T \mathcal{K}^t g(\boldsymbol{F}^\tau(\boldsymbol{x})) e^{-i\omega\tau} d\tau \\
=& \frac{1}{T} \int_{\tau=0}^T g(\boldsymbol{F}^{\tau+t}(\boldsymbol{x})) e^{-i\omega\tau} d\tau \\
=& \frac{1}{T} \int_{\tau=0}^T g(\boldsymbol{F}^\tau(\boldsymbol{x})) e^{-i\omega(\tau-t)} d\tau \\
=& e^{i\omega t} \frac{1}{T} \int_{\tau=t}^T g(\boldsymbol{F}^\tau(\boldsymbol{x})) e^{-i\omega\tau} d\tau.
\end{aligned}
\tag{C.19}
$$

Therefore, we have

$$
\begin{aligned}
|\mathcal{K}^t g_\omega(\boldsymbol{x}) - e^{i\omega t} g_\omega(\boldsymbol{x})| =& |e^{i\omega t} \frac{1}{T} \int_{\tau=t}^T g(\boldsymbol{F}^\tau(\boldsymbol{x})) e^{-i\omega\tau} d\tau - e^{i\omega t} \frac{1}{T} \int_{\tau=0}^T g(\boldsymbol{F}^\tau(\boldsymbol{x})) e^{-i\omega\tau} d\tau| \\
=& \frac{1}{T} |e^{i\omega t} \int_{\tau=0}^t g(\boldsymbol{F}^\tau(\boldsymbol{x})) e^{-i\omega\tau} d\tau| \\
\leq& \sqrt{\frac{2}{MT^2}} |e^{i\omega t} \int_{\tau=0}^t e^{-i\omega\tau} d\tau| \\
=& \sqrt{\frac{2}{MT^2}} |e^{i\omega t}(-i\omega(e^{-i\omega t} - 1)| \\
=& \sqrt{\frac{2}{MT^2}} |-i\omega + i\omega e^{i\omega t}| \\
\leq& \sqrt{\frac{2}{MT^2}} \Big[|i\omega| + |i\omega \cos\omega t| + |i\omega \sin\omega t|\Big] \\
\leq& \sqrt{\frac{2}{MT^2}} 3\omega_{\max} \leq \epsilon.
\end{aligned}
\tag{C.20}
$$

So, when $T \geq \sqrt{\frac{2}{M}} \frac{3\omega_{\max}}{\epsilon}$, $g_\omega(\boldsymbol{x})$ can approximate any Koopman eigenfunction with $\epsilon$ accuracy, for any $\epsilon > 0$

(ii) When $T \to \infty$, $g_\omega(\boldsymbol{x})$ can be defined as:

$$
g_\omega(\boldsymbol{x}) = \lim_{T \to \infty} \frac{1}{T} \int_{\tau=0}^T g(\boldsymbol{F}^\tau(\boldsymbol{x})) e^{-i\omega\tau} d\tau.
\tag{C.21}
$$

Let the Koopman operator $\mathcal{K}^t$ acts on $g_\omega(\boldsymbol{x})$, such that:

$$
\begin{aligned}
\mathcal{K}^t g_\omega(\boldsymbol{x}) =& \lim_{T \to \infty} \frac{1}{T} \int_{\tau=0}^T \mathcal{K}^t g(\boldsymbol{F}^\tau(\boldsymbol{x})) e^{-i\omega\tau} d\tau \\
=& \lim_{T \to \infty} \frac{1}{T} \int_{\tau=0}^T g(\boldsymbol{F}^{\tau+t}(\boldsymbol{x})) e^{-i\omega\tau} d\tau \\
=& \lim_{T \to \infty} \frac{1}{T} \int_{\tau=0}^T g(\boldsymbol{F}^\tau(\boldsymbol{x})) e^{-i\omega(\tau-t)} d\tau \\
=& e^{i\omega t} \lim_{T \to \infty} \frac{1}{T} \int_{\tau=t}^T g(\boldsymbol{F}^\tau(\boldsymbol{x})) e^{-i\omega\tau} d\tau \\
=& e^{i\omega t} \Big[\lim_{T \to \infty} \frac{1}{T} \int_{\tau=0}^T g(\boldsymbol{F}^\tau(\boldsymbol{x})) e^{-i\omega(\tau-t)} d\tau - \alpha\Big] \\
=& e^{i\omega t} \lim_{T \to \infty} \frac{1}{T} \int_{\tau=0}^T g(\boldsymbol{F}^\tau(\boldsymbol{x})) e^{-i\omega(\tau-t)} d\tau \\
=& e^{i\omega t} g_\omega(\boldsymbol{x}).
\end{aligned}
\tag{C.22}
$$

Therefore, when $T \to \infty$, $g_\omega$ is an eigenfunction of the Koopman operator $\mathcal{K}^t$. $\qquad\square$

# E   THE DETAILED PROCESS FOR TIME SERIES DECOMPOSITION

For the given time series, we first remove its global trends and seasonal effects. Such operations would cause the residual of the time series to tend to be stochastic fluctuation, which is the main attribution of the non-stationarity of real-world time series. Then, the detrended and deseasonalized residuals are divided into $J$ segments $\boldsymbol{X}_T^r = [\boldsymbol{x}_1^r, \boldsymbol{x}_2^r, \ldots, \boldsymbol{x}_J^r]$ and the index $S_v$ is calculated based on the Kolmogorov–Smirnov test to determine the proportion of the stationary and non-stationary components. After that, we perform the Fourier transform in the original segments $\boldsymbol{X}_T = [\boldsymbol{x}_1, \boldsymbol{x}_2, \ldots, \boldsymbol{x}_J]$ to calculate the frequency spectrum and sort all frequencies by the number of occurrences. Finally, the top $\alpha$ percent of the frequency spectrum are considered as the components of the stationary, while the remaining are considered as the components of the non-stationary. The disentanglement in the given time series $\boldsymbol{X}_T$ is mathematically formulated as follows:

$$\begin{aligned}
\boldsymbol{X}_s &= \mathrm{FT}^{-1}(S_\alpha, \mathrm{FT}(\boldsymbol{X}_T)), \\
\boldsymbol{X}_{ns} &= \mathrm{FT}^{-1}(S - S_\alpha, \mathrm{FT}(\boldsymbol{X}_T)), \\
\boldsymbol{X}_T &= \boldsymbol{X}_s + \boldsymbol{X}_{ns}
\end{aligned} \tag{D.1}$$

where $\boldsymbol{X}_s$, $\boldsymbol{X}_{ns}$ are time-invariant and time-variant components respectively. $S$ is the set of frequency spectrum. $S_\alpha$ is the set of global shared frequency spectrum. $\mathrm{FT}^{-1}$ denotes the inverse of FT.

In our work, we do not focus on the specific design for the detrending and deseasonalizing. Therefore, we conduct it by the commonly used Pytorch code with the additive model of the *seasonal_decompose* function. The additive model deems that the time series $\boldsymbol{X}$ consists of three components, including trend (i.e., the global trends) $\boldsymbol{X}_{trend}$, seasonal $\boldsymbol{X}_{trend}$, and residual $\boldsymbol{X}_r$. $\boldsymbol{X} = \boldsymbol{X}_{trend} + \boldsymbol{X}_{trend} + \boldsymbol{X}_r$. The *seasonal_decompose* function directly return the components, trend, seasonal, and residual in the code. More precisely, the flow of the *seasonal_decompose* function mainly includes four steps: (1) Determine the seasonal cycle (i.e., period) of the data. The period denotes the length of the season; (2) Compute the trend components. The seasonal component is the remaining cyclical pattern after removing the trend component; (3) Compute the trend components; (4) compute the residual by $\boldsymbol{X}_r = \boldsymbol{X} - \boldsymbol{X}_{trend} - \boldsymbol{X}_{trend}$.

# F   ADDITIONAL EXPERIMENTAL RESULTS

## F.1   ADDITIONAL RESULTS ON MULTIVARIATE TIME SERIES FORECASTING

Due to the limited pages, we list additional multivariate time series forecasting results. The results on **ETTh1**, **ETTm1**, and **ETTm2** datasets are reported in Table 5. As shown in Table 5, our KokerNet still achieves competitive performance compared with state-of-the-art deep forecasting models.

Table 5: Multivariate time series forecasting results with different forecasting lengths $H \in \{48, 96, 144, 192\}$ under the lookback length $T = 2H$ on **ETTh1**, **ETTm1**, and **ETTm2** datasets. The best results are highlighted in **bold** and the suboptimal results are highlighted in underline. (All the results of the compared methods are replications based on the publicly available code.)

| Models | | KokerNet | | Ns_Transformer | | Autoformer | | Koopa | | iTransformer | | KNF | | Crossformer | |
|---|---|---|---|---|---|---|---|---|---|---|---|---|---|---|---|
| Metric | | MSE | MAE | MSE | MAE | MSE | MAE | MSE | MAE | MSE | MAE | MSE | MAE | MSE | MAE |
| ETTh1 | 48 | **0.3366** | **0.3779** | 0.5152 | 0.4784 | 0.4722 | 0.4595 | 0.3455 | 0.3843 | 0.3442 | 0.3800 | 0.8760 | 0.7090 | 0.3545 | 0.3989 |
| | 96 | 0.4003 | 0.4177 | 0.5436 | 0.5064 | 0.5003 | 0.4746 | **0.3871** | **0.4058** | 0.3991 | 0.4150 | 0.9750 | 0.7440 | 0.4082 | 0.4258 |
| | 144 | **0.4068** | **0.4205** | 0.5473 | 0.5064 | 0.4670 | 0.4666 | 0.4298 | 0.4289 | 0.4165 | 0.4244 | 0.8010 | 0.6620 | 0.5002 | 0.4955 |
| | 192 | **0.4226** | **0.4352** | 0.6211 | 0.5287 | 0.5060 | 0.4802 | 0.4401 | 0.4357 | 0.4427 | 0.4503 | 0.9410 | 0.7440 | 0.5782 | 0.5645 |
| ETTm1 | 48 | **0.2942** | 0.3452 | 0.4084 | 0.4003 | 0.8157 | 0.5999 | 0.2863 | **0.3361** | 0.3162 | 0.3565 | 1.0260 | 0.7920 | 0.3128 | 0.3654 |
| | 96 | **0.2960** | **0.3452** | 0.4419 | 0.4429 | 0.5762 | 0.5124 | 0.3264 | 0.3648 | 0.3019 | 0.3483 | 0.9570 | 0.7820 | 0.3235 | 0.3670 |
| | 144 | **0.3163** | **0.3612** | 0.5081 | 0.4459 | 0.7313 | 0.5649 | 0.3546 | 0.3798 | 0.3216 | 0.3637 | 0.9210 | 0.7600 | 0.3667 | 0.4019 |
| | 192 | **0.3301** | **0.3719** | 0.5379 | 0.4661 | 0.6689 | 0.5421 | 0.3683 | 0.3875 | 0.3378 | 0.3760 | 0.8960 | 0.7310 | 0.3820 | 0.4213 |
| ETTm2 | 48 | **0.1396** | **0.2315** | 0.1726 | 0.2603 | 0.1919 | 0.2941 | 0.1403 | 0.2326 | 0.1415 | 0.2361 | 0.6210 | 0.6230 | 0.1860 | 0.2938 |
| | 96 | **0.1739** | **0.2576** | 0.2414 | 0.3092 | 0.2852 | 0.3530 | 0.1804 | 0.2614 | 0.1850 | 0.2736 | 1.5350 | 1.0120 | 0.3818 | 0.436 |
| | 144 | **0.2111** | 0.2871 | 0.3705 | 0.3827 | 0.2749 | 0.3453 | 0.2155 | **0.2859** | 0.2190 | 0.2971 | 1.3370 | 0.8760 | 0.4135 | 0.4796 |
| | 192 | **0.2301** | 0.3033 | 0.3237 | 0.3540 | 0.3039 | 0.3633 | 0.2401 | **0.3009** | 0.2393 | 0.3098 | 1.3550 | 0.9080 | 0.6551 | 0.6130 |

## F.2 THE INFLUENCE OF THE PROPORTION OF THE STATIONARY COMPONENTS AND SEGMENTATION LENGTH

To explore the influence of the proportion of the stationary components on the results and the influence of segmentation length on the results, we also include additional experiments. In Figure 5, we report the MSE results on **Traffic**, **ETTh1**, **ETTm1**, and **ETTm2**, which is not reported in the main paper due to the limited pages. From Figure 5, we can observe that the MSE tends to decrease as the proportion of stationary components increases.

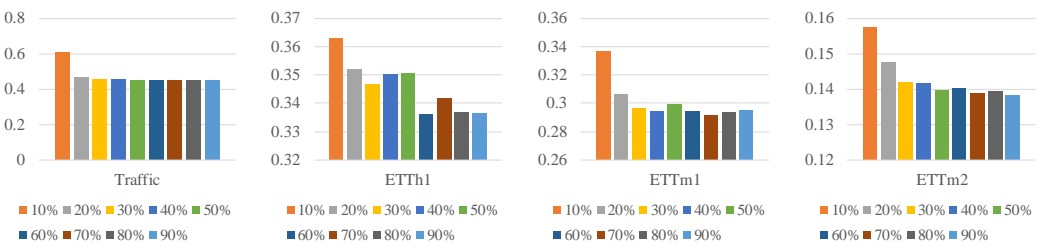

Figure 5: The influence of the proportion of the stationary components on the results.

In order to provide a more intuitive comparison of the influence of the stationary components, we analyze the result on the **M4** dataset again. Concretely, we first show the forecastability, Lyapunov exponents, trend, and seasonality of the M4 dataset with different frequencies in Table 6. ↓ indicates the smaller the value, the higher the non-stationarity, while ↑ indicates the higher the value, the higher the non-stationarity. Then, we report the forecasting results on the **M4** dataset (*i.e.,* the univariate time series forecasting results in the main text) in Table 7. From Table 6 and Table 7, we can observe that our KokerNet exhibits enhanced performance in scenarios where seasonality is less pronounced and forecastability is heightened. By contrast, our KokerNet tends to perform mediocre in scenarios where seasonality is more pronounced. In particular, our KokerNet achieves a 12.71% reduction in sMAPE for *Yearly* data and a 10.31% reduction for *Daily* data. This demonstrates that our proposal can perform better on datasets and yield superior results on datasets with higher non-stationarity.

Table 6: The forecastability, Lyapunov exponents, trend, and seasonality of the **M4** dataset with different frequencies. ↓ indicates the smaller the value, the higher the non-stationarity, while ↑ indicates the higher the value, the higher the non-stationarity. The results are cited from Wang et al. (2023).

|  | Forecastability (↓) | LEs (↑) | Trend (↓) | Seasonality (↓) |
|---|---|---|---|---|
| Yearly | 0.58 | 0.004 | 4.32 | 0.00% |
| Quarterly | 0.47 | 0.003 | 1.06 | 84.51% |
| Monthly | 0.44 | 0.011 | 0.48 | 66.34% |
| Weekly | 0.43 | 0.013 | 0.13 | 0.00% |
| Daily | 0.44 | 0.020 | 0.05 | 0.00% |
| Hourly | 0.46 | 0.003 | 0.02 | 99.76% |

In Figure 6, we report the MSE result with different forecasting length $H = \{48, 96, 144, 192\}$ on **ECL**. From Figure 6, we can observe that the time series tends to become more stationary as the length of the time series increases, and the proportion of stationary components has a greater impact on the results.

## F.3 TIME SERIES DECOMPOSITION

As previously discussed, the real-world time series commonly contains both time-invariant and time-variant patterns, corresponding to the stationary and non-stationary components. Therefore, we decompose the time series into these two components to evade information loss and the introduction of unnecessary disturbances caused by the single-component assumption. To evaluate the effectiveness of the time series decomposition step, we perform an experiment with three cases, including

Table 7: Univariate time series forecasting results with different frequencies on **M4** dataset. The best results are highlighted in **bold**. (All the results of the compared methods are replications based on the publicly available code.)

| | KokerNet | | | PatchTST | | | DLinear | | | Koopa | | |
|---|---|---|---|---|---|---|---|---|---|---|---|---|
| | sMAPE | MAPE | MASE | sMAPE | MAPE | MASE | sMAPE | MAPE | MASE | sMAPE | MAPE | MASE |
| Yearly | **13.454** | **16.571** | **3.033** | 16.668 | 23.302 | 3.729 | 15.413 | 18.467 | 3.696 | 14.707 | 19.417 | 3.275 |
| Quarterly | **10.213** | **11.779** | **1.192** | 12.606 | 15.118 | 1.628 | 10.546 | 12.288 | 1.242 | 10.775 | 12.823 | 1.287 |
| Monthly | **12.780** | **14.874** | **0.940** | 15.859 | 19.902 | 1.273 | 13.233 | 15.750 | 0.985 | 16.127 | 19.378 | 1.270 |
| Weekly | 11.157 | 10.309 | 3.396 | 11.551 | 11.234 | 4.465 | 11.168 | 12.003 | 5.936 | **10.221** | **9.542** | **3.135** |
| Daily | **3.035** | **4.387** | **3.251** | 3.576 | 5.590 | 3.894 | 3.384 | 5.165 | 3.685 | 3.395 | 4.886 | 3.682 |
| Hourly | 18.013 | 23.685 | 3.094 | 34.211 | 118.404 | 10.752 | **17.223** | **23.482** | **2.702** | 18.171 | 23.683 | 2.808 |
| Others | **4.858** | **6.410** | **3.248** | 6.685 | 15.336 | 4.503 | 5.089 | 7.173 | 3.765 | 5.109 | 6.777 | 3.570 |
| Average | **12.408** | **14.739** | **1.623** | 15.474 | 20.841 | 2.535 | 13.269 | 16.089 | 2.196 | 14.476 | 17.862 | 2.207 |

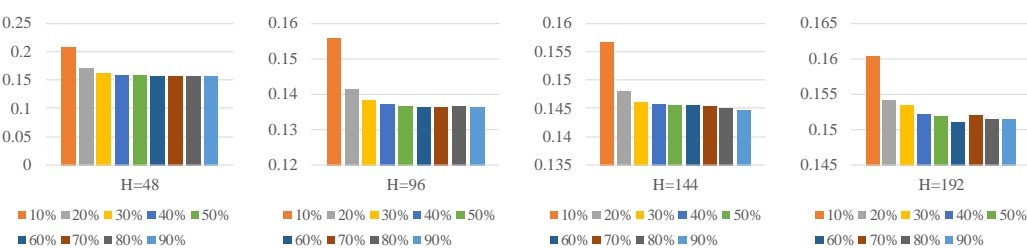

Figure 6: The influence of the segmentation length on the results.

$\mathcal{K}_s$, $\mathcal{K}_{ns}$, and $\mathrm{Pre}_{def}$. $\mathcal{K}_s$ denotes the entire time series is stationary, and only the global Koopman operator is learned for the time series. $\mathcal{K}_{ns}$ denotes the entire time series is non-stationary, and only the local Koopman operator is learned. Specifically, in the $\mathrm{Pre}_{def}$ case, we set the candidate range of the proportions as $[10\%, 20\%, \ldots, 90\%]$ and select the most optimal result, where $10\%$ represents that the stationary component account for $10\%$, while the non-stationary component account for $90\%$.

In the main text, we just report the results of two datasets. To verify the generalization effectiveness of the time series decomposition step, We also include the additional results on the remaining datasets of the main text. The results are shown in Table 8. We can observe that the results are consistent with our conclusions in the main text (*i.e.*, time series decomposition is important for the non-stationary data).

Table 8: The results with the single component. Here, $\mathcal{K}_s$ denotes only the global shared Koopman operator included in our model, $\mathcal{K}_{ns}$ denotes only local Koopman operator included in our model, and $\mathrm{Pre}_{def}$ denotes the best result for different proportion of the stationary component under the decomposition case. The best results are highlighted in **bold**.

| Dataset | | ETTh1 | | ETTm1 | | ETTm2 | | Traffic | | Weather | | Exchange | | ECL | |
|---|---|---|---|---|---|---|---|---|---|---|---|---|---|---|---|---|
| Metric | | MSE | MAE | MSE | MAE | MSE | MAE | MSE | MAE | MSE | MAE | MSE | MAE | MSE | MAE |
| $\mathcal{K}_s$ | 48 | 0.3368 | 0.3793 | **0.2899** | 0.3393 | 0.1385 | 0.2304 | 0.4475 | **0.2984** | **0.1389** | **0.1772** | **0.0427** | **0.1425** | 0.1570 | 0.2437 |
| | 96 | **0.3892** | **0.4118** | 0.2969 | 0.3461 | 0.1750 | 0.2591 | 0.4073 | 0.2827 | **0.1631** | **0.2091** | **0.0861** | **0.2071** | 0.1365 | 0.2291 |
| | 144 | 0.4112 | 0.4242 | 0.3169 | 0.3627 | **0.2073** | **0.2869** | 0.4123 | 0.2892 | **0.1813** | **0.2292** | **0.1360** | **0.2656** | 0.1453 | 0.2373 |
| | 192 | **0.4222** | 0.4357 | 0.3316 | 0.3726 | 0.2351 | 0.3073 | 0.4198 | 0.2947 | 0.2033 | 0.2516 | 0.2169 | 0.3369 | 0.152 | **0.2441** |
| $\mathcal{K}_{ns}$ | 48 | 0.6882 | 0.5521 | 0.6813 | 0.5412 | 0.1938 | 0.2867 | 1.3573 | 0.7819 | 0.1971 | 0.2580 | 0.0701 | 0.1868 | 0.8449 | 0.7630 |
| | 96 | 0.7035 | 0.5634 | 0.6021 | 0.5160 | 0.2210 | 0.3076 | 1.3794 | 0.7969 | 0.2338 | 0.2859 | 0.1633 | 0.2895 | 0.8326 | 0.7575 |
| | 144 | 0.7106 | 0.5783 | 0.6468 | 0.5378 | 0.2539 | 0.3290 | 1.3981 | 0.8058 | 0.2623 | 0.3092 | 0.2712 | 0.3823 | 0.8395 | 0.7596 |
| | 192 | 0.7239 | 0.5868 | 0.6052 | 0.5257 | 0.2859 | 0.3519 | 1.4088 | 0.8058 | 0.2862 | 0.3274 | 0.3666 | 0.4497 | 0.8479 | 0.7627 |
| $\mathrm{Pre}_{def}$ | 48 | **0.3363** | **0.3792** | 0.2916 | **0.3392** | **0.1383** | **0.2299** | 0.4458 | 0.2986 | 0.1398 | 0.1777 | 0.0452 | 0.1469 | **0.1564** | **0.2432** |
| | 96 | 0.4003 | 0.4177 | **0.2958** | **0.3451** | **0.1739** | **0.2576** | **0.4065** | **0.2816** | 0.1657 | 0.2125 | 0.0862 | 0.2074 | **0.1363** | **0.2285** |
| | 144 | **0.4064** | **0.4202** | 0.3163 | 0.3612 | 0.2109 | 0.2870 | 0.4089 | 0.2863 | 0.1819 | 0.2292 | 0.1389 | 0.2693 | **0.1446** | **0.2367** |
| | 192 | 0.4225 | **0.4351** | **0.3301** | **0.3719** | **0.2300** | **0.3034** | **0.4159** | **0.2914** | **0.2020** | **0.2495** | **0.2034** | **0.3261** | **0.1511** | 0.2442 |

## F.4 MODEL EFFICIENCY COMPARISON

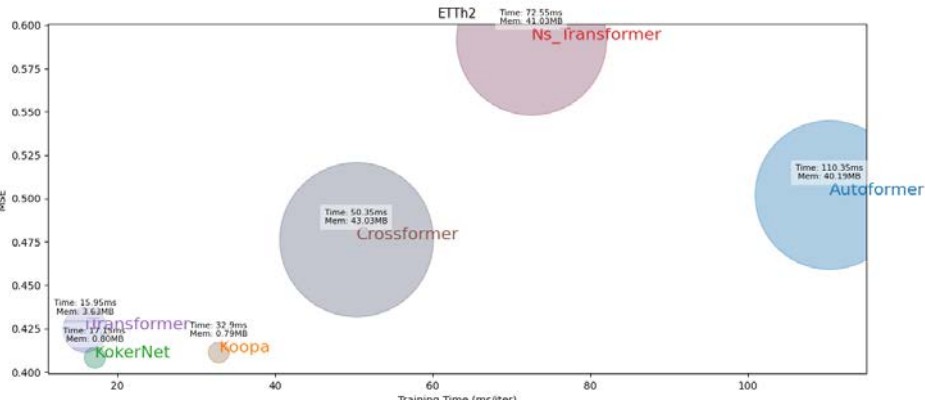

Figure 7: Model efficiency comparison on ETTh1 with $H = 144$. Training time and memory footprint are recorded with the same batch size (32) and official code configuration.

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
