# OpenReview forum: "KokerNet: Koopman Kernel Network for Time Series Forecasting"
_ICLR.cc/2025/Conference — Submitted to ICLR 2025_

### Official Review · Reviewer_jmbL · 2024-11-03

**Soundness:** 3
**Presentation:** 4
**Contribution:** 3
**Rating:** 6
**Confidence:** 4

**Summary:**

This paper introduces Koopman Kernel Network (KokerNet), a novel approach for time series forecasting that addresses the computational and interpretability challenges of existing Koopman-based methods. It uses a spectral kernel method to create a low-dimensional feature space, reducing computational costs. KokerNet decomposes time series into stationary and non-stationary components using a Kolmogorov-Smirnov (KS) test-based index, providing a more interpretable decomposition. It also includes a distribution module to handle time-varying distributions. Experiments show that KokerNet outperforms state-of-the-art models, enhancing both efficiency and interpretability.

**Strengths:**

1. The paper is overall well written

2. KokerNet enhances efficiency and interpretability by using a spectral kernel method for dimensionality reduction and decomposing time series into stationary and non-stationary components with a KS test-based index, outperforming state-of-the-art models.

3. The idea of integrating a distribution constraint into the forecasting process is sound.

**Weaknesses:**

The authors claim that decompositions in previous works often rely on empirical determinations of component composition and proportions, which lack interpretability. They propose to utilize the Kolmogorov-Smirnov (KS) test to guide the decomposition of time series into stationary and non-stationary components. I believe that a comparative analysis with these methods through ablation experiments is necessary. For instance, consider employing the approach based on Fourier transformation coefficients from reference [1] to differentiate between stationary and non-stationary components, replacing the KS test index selection in the proposed model.

[1] Koopa: Learning Non-stationary Time Series Dynamics with Koopman Predictors

**Questions:**

1. In the design of the measurement function, the encoder utilizes a mathematical function in the form of a Reproducing Kernel Hilbert Space (RKHS), while the decoder adopts a neural network in the form of a Multi-Layer Perceptron (MLP). What is the rationale behind this asymmetrical structure?

2. Regarding the segmentation of non-stationary components, what criteria are used to determine the segmentation length? Is the final performance sensitive to this parameter?

---

> ### Author Response · Authors · 2024-11-18
> **Response**
>
> Thank you for taking the time to our work. Your valuable comments are helpful in improving our work. We will provide a point-to-point response.
>
> **Q1: What is the rationale behind this asymmetrical structure?**
>
> **A1:** In practice, the decoder can be any form. In KokerNet, we select MLP. That is because (1) one of the main contributions of this paper is constructing the measurement function and learning the Koopman operator to capture the dynamic patterns of the time series, which does not have any special requirements for the decoder. Therefore, we tend to select a simple model as the decoder. (2) To better optimize the proposed model. In our KokerNet, the Koopman operator learning can be simply formulated as: $\hat{\bf{x}} = \Phi_ {de}(\mathcal{K}g(\bf{x}))$, where $\bf{x}$ and $\hat{\bf{x}}$ are the input and output, respectively. $g$ denotes the kernel mapping with RFF, $\mathcal{K}$ denotes the Koopman operator, and $\Phi_ {de}$ denotes a decoder. If the decoder is selected from the RKHS, the model can be considered as a stack of the RFF mapping. The RFF mapping can be seen as the single layer of the neural network with the cosine activation. The stack of the RFF mapping with the multilayer periodic function is prone to local minima. To avoid this case, we select MLP as the decoder for this paper.
>
> **Q2:  Ablation experiments in index**
>
> **A2:** For the time series decomposition, the process is that the Fourier transform is first performed in the original segments $\bf{X}_T = [\bf{x}_1, \bf{x}_2, \dots, \bf{x}_J ]$ to calculate the frequency spectrum and sort all frequencies by the number of occurrences. Then, the top \(\alpha\) percent of the frequency spectrum are considered as the components of the stationary, while the remaining are considered as the components of the non-stationary.
> The difference between our KokerNet and Koopa is how to determine the value of $\alpha$. In our method, index $S_v$ can guide us to determine the value of $\alpha$. By contrast, Koopa tends to empirically determine the value of $\alpha$.
>
> To evaluate the effect of the index $S_v$, we compare the performance under two cases, "index" and "emp", on **ETTh2** and **ILI** datasets. "index" denotes the result guided by $S_v$. "emp" denotes the empirical case and shows the best result for different proportions of the stationary component under the decomposition case. In the "emp" case, we set the candidate range of the proportions as [10\%, 20\%,..., 90\%] and select the most optimal result, where 10\% represents that the stationary component accounts for 10\%, while the non-stationary component accounts for 90\%. All the results are reported in the following Table. For **ETTh2** data, $S_v^{\text{ETTh2}} = 0.0865$, indicating that the proportion of non-stationary components is minimal and it does not need to conduct the decomposition. The result is better than the best result of the "emp" case. For the **ILI** dataset, $S_v^{\text{ILI}} = 0.2653$, meaning that it includes more non-stationary components. We will decompose the time series with $\alpha=30$. The result is equal to the best result of the "emp" case. To sum up, the index can effectively guide us to conduct the decomposition and lead to a better result.
> ||||**index**||||**emp**|||
> |-|-|:-:|:-:|:-:|:-:|:-:|:-:|:-:|:-:|
> ||| 48(24)|96(36)|144(48)|192(60)|48(24)|96(36)|144(48)|192(60)|
> |-|-|:-:|:-:|:-:|:-:|:-:|:-:|:-:|:-:|
> |**ETTh2**|MSE|**0.2299**|**0.2929**|**0.3277**|**0.3575**|0.2303|0.2960|0.3336|0.3579|
> || MAE | **0.3036** |**0.3467**|**0.3727**|**0.3929**|0.3043|0.3499|0.3766|**0.3929**|
> |-|-|:-:|:-:|:-:|:-:|:-:|:-:|:-:|:-:|
> |**ILI**|MSE|**1.8710**|**1.9181**|**1.8849**|**1.9347**|**1.8710**|**1.9181**|**1.8849**|**1.9347**|
> || MAE |**0.8351**|**0.8934**|**0.9223**|**0.9583**|**0.8351**|**0.8934**|**0.9223**|**0.9583**|
>
> **Q3: Regarding the segmentation of non-stationary components, what criteria are used to determine the segmentation length? Is the final performance sensitive to this parameter?**
>
> **A3:** In our paper, the segmentation length is set as 48, the minimum forecasting length. In order to test the influence of the segmentation length on the performance, we compute the index under different segmentation lengths on **ETTh2** and **Exchange** datasets. The results are shown in the following Table. As shown in the table, the segment length has a slight impact on the index $S_v$ but does not affect the configuration of $\alpha$. Therefore, the final performance is not sensitive to the segmentation length. In the revised manuscript, we will include it to analyze parameter sensitivity.
> |||**Segmentation length**|||
> |-|:-:|:-:|:-:|:-:|
> ||48($\alpha$)|96($\alpha$)|144 ($\alpha$)|192 ($\alpha$)|
> |-|:-:|:-:|:-:|:-:|
> |**ETTh2**|0.0865(100)| 0.0633(100)|0.0537(100)|0.0513(100)|
> |**Exchange**|0.1817(80)|0.1778(80)|0.1703(80)|0.1667(80)|
>
> **If you have any further questions, we will discuss them in a timely manner. That could help improve our work.**

---

> ### Comment · Reviewer_jmbL · 2024-11-27
>
> Thank you for addressing my concerns! I'd like to increase the confidence score.

---

> > ### Author Response · Authors · 2024-11-29
> > **Response**
> >
> > Thank you for taking the time to read our rebuttal carefully and replying. We are also pleased to hear that our clarifications can address your concerns and receive positive feedback. If you have any further questions, we will address them in a timely manner. This will help improve our work. We again appreciate your feedback.

---

### Official Review · Reviewer_4P4B · 2024-11-04

**Soundness:** 3
**Presentation:** 3
**Contribution:** 2
**Rating:** 3
**Confidence:** 4

**Summary:**

The paper proposes learning the Koopman operator in a proposed low dimensional space by using cosine functions, motivated by a use of Bochner's theorem and kernel integral transforms. It performs several studies in relation to time series forecasting to assess the merits of the proposed methodology.

**Strengths:**

The paper showcases an innovative development at the intersection of (1) Kernel integral transforms theory, (2) Koopman operator theory. An interesting decomposition is presented in order to tackle the non-stationarity of the time series which can appear in practical settings.

**Weaknesses:**

The majority of the paper doesn't present much perceivable novelty in my opinion. For example there is an attempt in Appendix C.1, Lemma C.1 to *prove* an obvious application of Bochner's theorem which anyway has already been done in the famous "Random Fourier Features" paper. This is already known several times over and thus there should be a *compelling* reason to present its proof and make it appear like its your own innovation (I am not against re-writing well-known proofs but it must be for a good reason). There is no apparent reason in the paper why this should be shown as a line-by-line proof and this leads me to the suspicion of there being an artificial mathematical padding of space, unfortunately, in order to try and obfuscate the reader.

This is amplified when considering Lemma C.2 and Theorem C.1, since the Chernoff-based exponential probability bound (C.7) proven doesn't appear to serve any intrinsic purpose and appears to be a standard matrix probability theorem slightly re-arranged so as to obfuscate the readability of the paper. Such bounds *are* useful don't get me wrong, but usually then there needs to be a stronger commentary and appeal to something like Computational Learning Theory. One can argue that it is useful in a sense for then proving Theorem C.1 in particular for equation C.16 but then ultimately what does Theorem C.1 intend to say? That ultimately with a high enough number of samples there is a convergence between a kernel approximator and a mean estimator of a set of random matrices? This is quite an obvious conclusion and doesn't need 1+ pages of proof. Of course it *can* be interesting if the bound intends to be studied in some sufficient manner (as per Computational Learning Theory), but then it is not made any reference to in the actual paper other than "here is a complex bound". An appeal to Law of Large Numbers in a random matrix form would be much more simpler, familiar, and appropriate than what is currently presented for example, just to say "it works" rather than "how it works" according to sampling number  complexity

And for example Theorem C.2 (ii) is something that should be known to the reader *if* they are quite familiar with Koopman Operator Theory in practice and thus no proof is really required yet one is presented which essentially overlaps the intended purpose of the previous proofs / lemmas, which ultimately all (over the course of 3+ pages) serves to raise the point "it works in large sample rates", which unless there is good reason to belief this should not be the case, is an obvious conclusion. Ultimately to a reader not well versed in mathematics this would appear to do nothing but introduce of a lot of new notation to intentionally obfuscate the readability of the paper, in my opinion.

Mathematics aside, there does not appear to be ample quantitative results to show that the proposed method actually works out in practice and thus should be used as an alternative to other competing Koopman operator based methods. For example for the purpose of forecasting why should someone be motivated to use this method over the standard DMD-based methods which have been shown to be enjoying absolute plethora of use cases in the engineering fields which experience massive levels of non-stationarity and large dimensionality? The authors are suggested to read for example "Higher Order Dynamic Mode Decomposition and Its Applications". If the proposed methodology is to be used seriously as good alternative then it would be worth explicitly analyzing why it might do better than higher order dynamic mode decomposition and such, which are currently actively used in everyday engineering industries for problems such as spatio-temporal fluid dynamics *without* an explicit need for training. Essentially,  because "Koopman" is the name of methodology I would expect much more "Koopman" based comparisons and analysis.

**Questions:**

Since the purpose of the questions is to try and alter the reviewer's mind, it would be good if you could address those main claims raised in the "weakness" section if possible. In particular,

 - Why did you feel it was required to show a half-page proof of Bochner's Theorem, and do you feel that there is strong value in presenting a Chernoff-based bound of the methodology given that the bound is not really serving much purpose (to my own opinion), and that it leads more to the obfuscation of the text rather raise its intrinsic value?

 - How come you didn't try to compare the proposed methodology to other Kooopman operator based forecast methods that are actively used in industry?

---

> ### Author Response · Authors · 2024-11-18
> **Response**
>
> **Q1: Regarding the theorems involved in our paper**
>
> **A1:** Thanks for your insightful components, which are very helpful in improving the simplicity and readability of our paper. We will distill the theoretical framework, showcasing only the core insights in the revised manuscript. In the following, we will explain why the current theoretical framework is adopted.
>
> **Lemma C.1**  In our work, one of the main contributions is that we tend to approximate the implicit kernel mapping by random Fourier feature (i.e., Eq. (3) in the main text), enabling learning the Koopman operators in a low-dimensional feature space. Taking into account the diversity of the reader, we present a detailed derivation of Eq. (3) in Lemma C.1.
>
> **Theorem C.1** In our work, we consider the implicit kernel mapping to be the measurement function $f$ in the Koopman framework $\mathcal{K}f(\bf{s}_ t) = f(\bf{s}_ {t+1})$ to map the data into a sufficiently high-dimensional feature space. Furthermore, we approximate the implicit kernel mapping by random Fourier feature, enabling learning the Koopman operators in a low-dimensional feature space. Theorem C.1 shows that when the sampling number meets certain conditions, the low-dimensional mapping can approach the high-dimensional mapping well. On the one hand, it shows that low-dimensional mapping can approximate the effect of high-dimensional mapping, providing strong theoretical support for our method. On the other hand, in addition to the performance, one also wants to know how many random features should be selected, such as the reviewer **u2dN**. Theorem C.1 indirectly provides a theoretical guide for us to choose the number of random features. As a result, we include Theorem C.1 in our work to highlight the interpretability theoretically.
>
> **Theorem C.2 (ii)** We have meticulously reviewed Theorem C.2, especially Theorem C.2 (ii), and agree with your comment. We will distill (ii) in the revised manuscript to ensure the readability of our paper.

---

> ### Author Response · Authors · 2024-11-18
> **Response**
>
> **Q2: Regarding the standard DMD-based methods**
>
> **A2:** We have briefly read the book you recommended. It primarily introduces two methods (HODMD and STKD) and their applications across various fields, while also recognizing their effectiveness in system analysis and forecasting. In the following, we will show the differences between the standard DMD-based methods and our KokerNet and the potential benefits of KokerNet over the DMD-based methods.
>
> **Difference** (1) The way of learning. DMD-based methods are analysis methods based on data decomposition and do not require training. The main operation in DMD-based methods is matrix decomposition (singular value decomposition (SVD) or eigenvalue decomposition (EVD)). By contrast, our KokerNet is a neural network-based approach and requires training. Concretely, KokerNet can be formulated as: $\hat{\bf{x}} = \Phi_ {de}(\mathcal{K}g(\bf{x}))$, where $\bf{x}$ and $\hat{\bf{x}}$ are the input and output, respectively. $g$ denotes the kernel mapping with RFF, $\mathcal{K}$ denotes the Koopman operator, and $\Phi_ {de}$ denotes a decoder. We make $\mathcal{K}(\cdot)$ as the linear mapping, and thus $\mathcal{K}g(\bf{x})$ can be considered as a kernel network, stacking the non-linear kernel mapping with linear mapping. (2) We introduce a distribution module. A fundamental challenge with non-stationary time series is the time-varying distribution. Therefore, we introduce a distribution module to keep the forecasting to the distribution law of the time series or the system.
>
> **Benefits**  The main benefit of KokerNet over the DMD-based methods is that it reduces computational complexity, especially for large-scale problems. For DMD-based methods, SVD or EVD is commonly used, resulting in $\mathcal{O}(n^3)$ computational complexity. In contrast, our KokerNet involves only matrix multiplication and has $\mathcal{O}(nM)$ computational complexity, where $M \ll n$ is the sampling number in our paper.
>
> Due to the difference between DMD-based methods and our KokerNet, discussed above, we just touched on the DMD-based methods a little bit in the introduction section, and do not perform the comparison with the DMD-based methods. Note that we have compared with the other Koopman operator-based methods (Koopa and KNF) that focus on learning the Koopman operator within the framework of neural networks. To include more "Koopman" based comparisons, we have added another two Koopman-based methods on **Exchange** and **ECL** datasets with the forecasting length $H=96,192$. The results are reported in the following Table. The results show that our KokerNet is comparable in most cases.
> |  | Models | KokerNet | Koopa [1] | KNF [2] | Koopformer [3] | Koopman-Wavelet [4] |
> | --- | --- | --- | --- | --- | --- | --- |
> | Metric  || MSE | MSE | MSE | MSE | MSE |
> | Exchange | 96 | $\underline{0.0897}$ | 0.0916 | 0.2940 | 0.1360 | **0.0849** |
> | Exchange | 192 | **0.1862** | $\underline{0.1892}$ | 0.6540 | 0.2750 | 0.1980 |
> | ECL | 96 | $\underline{0.1365}$ | 0.1389 | 0.1980 | 0.1870 | **0.1350** |
> | ECL | 192 | **0.1520** | 0.1566 | 0.2450 | 0.1920 | 0.1530 |
>
> [1] Yong Liu, Chenyu Li, Jianmin Wang, and Mingsheng Long. Koopa: Learning non-stationary time
> series dynamics with koopman predictors. In Advances in Neural Information Processing Sys-
> tems, volume 36, pp. 12271–12290, 2023
>
> [2] Rui Wang, Yihe Dong, Sercan O. Arik, and Rose Yu. Koopman neural operator forecaster for
> time-series with temporal distributional shifts. In Proceeding of the International Conference on
> Learning Representations, 2023
>
> [3] Hui Wang, Liping Wang, Qicang Qiu, Hao Jin, Yuyan Gao, Yanjie Lu, Haisen Wang, and Wei Wu.
> Koopformer: Robust multivariate long-term prediction via mixed koopman neural operator and
> spatial-temporal transformer. In Proceedings of the International Joint Conference on Neural
> Networks, pp. 01–08, 2023.
>
> [4] Liu Fu, Meng Ma, and Zhi Zhai. Deep koopman predictors for anomaly detection of complex iot
> systems with time series data. IEEE Internet of Things Journal, 2024.
>
> **If you have any further questions, we will discuss them in a timely manner. That could help improve our work.**

---

> ### Comment · Reviewer_4P4B · 2024-11-25
>
> > The main benefit of KokerNet over the DMD-based methods is that it reduces computational complexity,
>
> Yes I can agree that that is a benefit. But it is hard to find it a genuinely important contribution. Most time series datasets that are worked with in practice are really quite small (especially in comparison to the huge amount data and parameters we consider in relation to langauge models, image data sets etc..). Time series problems hardly approach anything even close to that level. It is often much better to then take the O(N^3) complexity, work with a smaller overall cardinality of bespoke data that truly represents the dynamics of the underlying system, and then (as you have noticed / pointed), even *without* the need for training then learn the dynamics of the signal up to an incredibly high degree of accuracy.
>
> Essentially I am coming to this from a pragmatic angle as if I were to make a recommendation to a friend / colleague / industry. In your title the word "Koopman" is used and we are aware that there a plethora of finite approximation Koopman based solutions for forecasting that have a large amount of convincing mathematical structure behind them, and which require absolutely minimal tuning for great results. I cannot forsee a situation where I would recommend using your model unfortunately over standard Koopman approximation principles already actively used in the econometrics fields, fluid flow fields, structural engineering fields (so we know they work! And for data situations much, much, much harder than what is typically explored in standard ML data set libraries in many cases).
>
> If you could have tackled this problem head on and gave a convincing argument as to why your method is objectively better for certain (realistic) tasks in industry as opposed to other competing Koopman based I would most definitely reconsider, and *that* would peak further curiousity from myself. Unfortunately I dont see this being done here, little more than an argument that goes beyond "sure something like HODMD is successful, doesn't require much data, doesn't even require parameter training .... but at least we're linear". I don't believe this is the correct angle to take when working with time series data. Developing a spatio-temporal dataset of a wind tunnel fluid flow can cost upwards of  1-10M dollars. This field cannot be treated as other fields in ML where data feels endless, so the label of "big data" rings trues, and the linearity would be greatly appreciated.
>
> > Due to the difference between DMD-based methods and our KokerNet, discussed above, we just touched on the DMD-based methods a little bit in the introduction section, and do not perform the comparison with the DMD-based methods.
>
> And there-in lies a major point. This is an important difference to comment on and address, which are you not addressing. In industry and pragmatic settings one situation is almost always favored over the other, as I re-iterate in many cases one does not come across the large data cases in time series anywhere near as those problems which can be solved via conventional DMD approaches used in the sciences and industry. In the future this point must be addressed and not simply given a passing mention.
>
> If you want a reference for a paper I feel actually brings a lot of justice to the combination of Deep Learning and Koopman Operators without neglecting the DMD roots (which are practically what we work with in a finite world), it would be "Deep learning for universal linear embeddingsof nonlinear dynamics" by Lusch et al. As you can see in the paper a lot of effort has gone into understanding the fundemental physics of the issue which is where / how such operators are practically used a lot.

---

> ### Comment · Reviewer_4P4B · 2024-11-25
>
> I appreciate taking my comments positively in this light (regarding to the comments on superfluous mathematics etc..). I do genuinely believe they will improve the paper for the better. Much appreciated.

---

> ### Author Response · Authors · 2024-11-29
> **Response**
>
> Thank you for taking the time to read our rebuttal carefully
> and additional comments. DMD-based approaches appear closely related to our KokerNet, we will include more DMD-based approaches in the revised manuscript. In the following, we show the reason why your method is objectively better for certain (realistic) tasks and recommend it to others. In addition, we also compare our method with DMD and eDMD methods on the small datasets.
>
> First, there is also a convincing mathematical structure behind our method. We model the temporal dependence of the time series as a kernel $ k(\bf{x}_ t, \bf{x}_ {t+1}) = \langle f(\bf{x}_ t), f(\bf{x}_ {t+1}) \rangle_ {\mathcal{H}}$. Defining the integral operator $\mathcal{P}$ as $\mathcal{P}f(\bf{x}_ t) = \int k(\bf{x}_ t, \bf{x}_ {t+1})  f(\bf{x}_ {t+1})\mu(d(\bf{x}_ {t+1}))$. We have $k(\bf{x}_ t, \bf{x}_ {t+1}) = \sum_ {m=1}^M \lambda_ m \varphi_ m(\bf{x}_ t)\varphi_ m(\bf{x}_ {t+1})$, where $\lambda_ m$ is the eigenvalue and $\varphi_ m$ is the eigenfunction of the operator $\mathcal{P}$. It is known that when $M \to \infty$ the operator $\mathcal{P}$ commutes with the Koopman operator $\mathcal{K}$. An important fact here is that the space spanned by eigenfunctions is the same for commuting operators. Therefore, we approximate the eigenfunctions of the Koopman operator by approximating the kernel $ k(\bf{x}_ t, \bf{x}_ {t+1})$ with the $\varphi_ m, m=1,\cdots$ (i.e., learning $g$ by the kernel network). Our theorem also proves this point.
>
>
> Second, we are not limited to any one field (such as your concerns in industry, which may have a small dataset). In practical applications, there are also many forecasting tasks with large datasets, such as traffic and weather prediction. The data for these tasks is relatively easy to collect and sufficient. In these tasks, high computational complexity is unacceptable. In addition, when data is collected, distribution drift can occur for various reasons (e.g., heat generated by long-running equipment), which is also a crucial issue to consider, even for small data. As previously discussed, we introduce a distribution constraint module to tackle this issue.
>
> Existing experiments show the superiority of our method in the large-scale dataset. To further demonstrate the performance of the proposal on the small dataset, we compare it with DMD and eDMD (The code of DMD and eDMD is from the public PyDMD codebase in GitHub). Since we do not have a small dataset that you are talking about in the industry, we take the first 500 points in time that we already have (ETTh1, ETTh2, ETTm1, ETTm2) as a standalone small data set to evaluate our method under $H=48$. The results (MSE) are shown in the following table and demonstrate that our KokerNet is superior to the DMD methods.
>
> |           | **ETTh1\_500** | **ETTh2\_500** | **ETTm1\_500** | **ETTm2\_500** |
> |-----------|----------------|----------------|----------------|----------------|
> | **DMD**   | 3.0720         | 7.8345         | 1.2155         | 2.6091         |
> | **eDMD**  | 3.8089         | 6.2291         | 1.0624         | 3.5871         |
> | **KokerNet** | **0.7682**    | **0.5754**     | **0.3716**     | **0.5381**     |
>
>
> **If you have any further questions, we will discuss them in a timely manner. That could help improve our work.**

---

> > ### Author Response · Authors · 2024-12-02
> > **Response**
> >
> > Dear Reviewer 4P4B,
> >
> > Apologize for our continuous messages. Since the end of the discussion period is approaching, we would appreciate if you could let us know whether the above answer and additional experiment address your comments or not and let us know if you have additional questions, comments, and concerns related to these points. Thanks again. In addition, we have also updated the comparison for the DMD-based methods. The results are updated in the following table.
> >
> > |                           | **ETTh1_500** | **ETTh2_500** | **ETTm1_500** | **ETTm2_500** |
> > |---------------------------|---------------|---------------|---------------|---------------|
> > | **DMD**                   | 3.0720        | 7.8345        | 1.2155        | 2.6091        |
> > | **eDMD**                  | 3.8089        | 6.2291        | 1.0624        | 3.5871        |
> > | **KRR[1]**   | 2.5595        | 13.3960       | 2.1210        | 7.2693        |
> > | **PCR[1]**   | 3.9560        | 12.7278       | 3.2041        | 8.0007        |
> > | **sKAF [2]**              | 2.3596        | 15.9180       | 0.8154        | 10.7259       |
> > | **KokerNet**              | **0.7682**    | **0.5754**    | **0.3716**    | **0.5381**    |
> >
> >
> > [1]  Meanti, G., Chatalic, A., Kostic, V., Novelli, P., Pontil, M., & Rosasco, L. (2024). Estimating Koopman operators with sketching to provably learn large scale dynamical systems. Advances in Neural Information Processing Systems, 36.
> >
> > [2]  Dimitrios Giannakis, Amelia Henriksen, Joel A. Tropp, and Rachel Ward. Learning to forecast dynamical systems from streaming data. SIAM Journal on Applied Dynamical Systems, 22(2): 527–558, 2023

---

### Official Review · Reviewer_SgkE · 2024-11-04

**Soundness:** 3
**Presentation:** 2
**Contribution:** 3
**Rating:** 6
**Confidence:** 4

**Summary:**

This paper presents KokerNet, a Koopman kernel network for time series forecasting. It addresses critical issues in existing methods, such as the high computational costs and the challenges posed by data distribution variations. By employing spectral kernel methods to construct a measurement function space, the authors achieve a notable reduction in computational burden. Furthermore, the decomposition of time series into stationary and non-stationary components is used to enhance interpretability. The global and local Koopman operators are effectively utilized to predict future behaviors of these components.

**Strengths:**

The application of Koopman operators for time series forecasting represents a significant advancement. The spectral kernel method for measurement functions is a commendable innovation that simplifies the computational process. The creation of an index to facilitate the decomposition into stationary and non-stationary components is particularly valuable. This not only enhances model interpretability but also provides insights into the dynamics of the data. Empirical results show KokerNet's superiority, underscoring its practicality and effectiveness in real-world applications.

**Weaknesses:**

The manuscript does not clearly explain the integration of the kernel method within the Koopman network framework. Readers are left without a clear understanding of how the spectral kernel functions operate within the Koopman operator or how these methods combine to provide computational advantages. More detailed explanations and illustrative diagrams would significantly aid comprehension and convey the novelty of the approach.

The manuscript also lacks citations and discussion of closely related and classical methods, particularly multiresolution DMD, which already employs a similar architecture for handling non-stationary time series. KokerNet’s model appears to be a special case of this approach. Expanding the literature review and explicitly contrasting KokerNet with multiresolution DMD would clarify the unique aspects of the method.

The theoretical components presented are largely standard and do not demonstrate clear, novel benefits for KokerNet’s performance or interpretability. The paper would be strengthened by a clearer, more thorough explanation of how the theory specifically benefits the model’s forecasting ability or contributes new insights to the field.

**Questions:**

1. Could the authors provide more detail on how the spectral kernel method integrates within the Koopman framework and specifically outline its role in enhancing efficiency?
2. Why is multiresolution DMD, which appears closely related, not discussed as part of the literature? Could the authors clarify the differences and potential benefits of KokerNet over multiresolution DMD?
3. Could the authors include more insights into how they ensured their model's robustness to real-world shifts in data distribution, especially for rapidly changing non-stationary components?

---

> ### Author Response · Authors · 2024-11-18
> **Response**
>
> Thanks for your valuable comments, which have helped improve our work. We will provide a point-to-point response in the rebuttal.
>
> **Q1: Could the authors provide more detail on how the spectral kernel method integrates within the Koopman framework and specifically outline its role in enhancing efficiency?**
>
> **A1:** The core idea of the Koopman framework is to characterize the complicated evolution via an infinite-dimensional linear Koopman operator by acting on the measurement function. It is formulated as follows:
> $$
> \mathcal{K}f(\bf{s}_ t) = f (\bf{s}_ {t+1})
> $$
> where $\mathcal{K}$ is the Koopman operator, $f: \mathbb{R}^d \to \mathbb{R}^D, D \to \infty$ is the measurement function.
>
> Intuitively, the Koopman framework tends to map the data into an infinite-dimensional feature space by $f$ and characterize the non-linear complicated evolution in a linear manner by $\mathcal{K}$ within the feature space. The kernel method has been proven to be a promising method to address the non-linear problem, and it can map the data into an infinite-dimensional feature space (i.e., RKHS) via implicit kernel mapping. Therefore, we consider the implicit kernel mapping to be the measurement function $f$. Furthermore, the temporal dependence can be modeled by the kernel. In our KokerNet, we approximate the implicit kernel mapping by a low-dimensional feature mapping $g$ based on Bochner’s theorem (i.e., Eq. (3) in the main text). As a result, we integrate the spectral kernel within the Koopman framework.
>
> Ideally, we want to map the data to a sufficiently high-dimensional feature space to linearize the complex evolution of a dynamic system. However, this would be computationally expensive. In our KokerNet, we approximate the implicit high-dimensional kernel mapping by a low-dimensional feature mapping $g$ based on Bochner’s theorem (i.e., Eq. (3) in the main text). **Theorem 3.1** shows that when the number of samples meets certain conditions, the low-dimensional mapping can approach the high-dimensional mapping well. This theorem not only shows the efficiency (low-dimensional mapping can approximate the effect of high-dimensional mapping, and thus reduce the compute cost) but also provides a theoretical guide for us to choose the number of random features. In addition, we compared the model efficiency in three aspects, including forecasting performance, training time, and memory footprint in section **4.3**. The results are reported in **Appendix E.4**, showing that our KokerNet has better forecasting performance with less training time and memory footprint.
>
> **Q2:  Why is multiresolution DMD, which appears closely related, not discussed as part of the literature? Could the authors clarify the differences and potential benefits of KokerNet over multiresolution DMD?**
>
> **A2:** The differences between multiresolution DMD and our KokerNet are: (1) The way of learning. Multiresolution DMD is an analysis method based on data decomposition and does not require training. The main operation in multiresolution DMD is matrix decomposition. By contrast, our KokerNet is a neural network-based approach and requires training. Concretely, KokerNet can be formulated as: $\hat{\bf{x}} = \Phi_ {de}(\mathcal{K}g(\bf{x}))$, where $\bf{x}$ and $\hat{\bf{x}}$ are the input and output, respectively. $g$ denotes the kernel mapping with RFF, $\mathcal{K}$ denotes the Koopman operator, and $\Phi_ {de}$ denotes a decoder. We make $\mathcal{K}(\cdot)$ as the linear mapping, and thus $\mathcal{K}g(\bf{x})$ can be considered as a kernel network, stacking the non-linear kernel mapping with linear mapping. (2) We introduce a distribution module. A fundamental challenge with non-stationary time series is the time-varying distribution. Therefore, we introduce a distribution module to keep the forecasting to the distribution law of the time series or the system.
>
> The main benefit of KokerNet over the multiresolution DMD is that it reduces computational complexity. For multiresolution DMD, singular value decomposition (SVD) or eigenvalue decomposition (EVD) is commonly used, resulting in $\mathcal{O}(n^3)$ computational complexity. In contrast, our KokerNet involves only matrix multiplication and has $\mathcal{O}(nM)$ computational complexity, where $M \ll n$ is the sampling number in our paper.
>
> We admit that multiresolution DMD appears closely related to our KokerNet. Due to the difference, discussed above (1), we just touched on the DMD-based method a little bit in the introduction section. We will include more multiresolution DMD approaches in the revised related works.

---

> ### Author Response · Authors · 2024-11-18
> **Response**
>
> **Q3: Could the authors include more insights into how they ensured their model's robustness to real-world shifts in data distribution, especially for rapidly changing non-stationary components?**
>
> **A3:**  On the one hand, the kernel method has been proven to be a promising method to address the non-linear problem, and it can map the data into an infinite-dimensional feature space (i.e., RKHS) via implicit kernel mapping. We consider the implicit kernel mapping to be the measurement function $f$. **Theorem 3.1** shows that when the number of samples meets certain conditions, the low-dimensional mapping can approach the high-dimensional mapping well. This theorem ensures that low-dimensional mapping in our KokerNet can approximate the effect of high-dimensional mapping. On the other hand, **Therome 3.2** shows that the function, induced by $g$, can approximate the eigenfunction of the Koopman operator, which means that our KokerNet at the encoding stage can reveal the dynamic patterns of the time series or system.
>
> In addition, we introduce a distribution module in our KokerNet. This module, especially for the non-stationary component, can keep the forecasting to the distribution law of the time series. We minimize the distribution loss $\mathcal{L}_ {\text{dis}}^{\text{ns}} = \mathcal{L}(\mathcal{N}(\bf{\mu}_ {\text{ns}}^i, \bf{\delta}_ {\text{ns}}^{2, i}), \mathcal{N}(\bf{\hat{\mu}}_ {\text{ns}}^i, \bf{\hat{\delta}}_ {\text{ns}}^{2,i})), i=2, \cdots, Q$, where $ \mathcal{N}(\bf{\mu}_ {\text{ns}}^i, \bf{\delta}_ {\text{ns}}^{2, i})$ is the ground truth, and $\mathcal{N}(\bf{\hat{\mu}}_ {\text{ns}}^i, \bf{\hat{\delta}}_ {\text{ns}}^{2,i})$ is the prediction based on the $\mathcal{K}_ {\text{dis}}$. Furthermore, we align the predicted distribution with the distribution of forecasts by minimizing the loss $\mathcal{L}_ {\text{dis}}^{\text{alig}} = \mathcal{L}(\mathcal{N}(\hat{\bf{\mu}}_ i, \hat{\bf{\delta}}_ i^2), \mathcal{N}(\bf{\hat{\mu}}_ {\text{ns}}^i, \bf{\hat{\delta}}_ {\text{ns}}^{2,i}))$. Therefore, for the rapidly changing non-stationary, our KokerNet not only captures the evolution of the value of the data using Koopman operator learning module but also explores the dynamics of the distribution utilizing the distribution constraint module, enabling the prediction to follow the distribution patterns of the time series.
>
> Furthermore, we also evaluate the performance of our KokerNet under different degrees of non-stationary. The non-stationary is reported in Table 6 and the performance is reported in Table 7 in **Appendix E.2**. All the results show that our KokerNet consistently outperforms the state-of-the-art models.
>
> **If you have any further questions, we will discuss them in a timely manner. That could help improve our work.**

---

> ### Comment · Reviewer_SgkE · 2024-11-30
>
> Thanks for your feedback. Basically, I can understand the contribution and organization of this paper more clearly now. Thus, I will keep the original rating.

---

> > ### Author Response · Authors · 2024-12-01
> > **Response**
> >
> > Thank you for taking the time to read our rebuttal carefully and reply. We are also pleased to hear that our clarifications can address your concerns and receive positive feedback. Thanks again.

---

> ### Author Response · Authors · 2024-12-02
> **Response**
>
> Dear Reviewer SgkE,
>
> We just want to let you know that we updated the comparison for the DMD-based methods. We think this modification is related to your comments about the DMD. Thank you again for your time and valuable comments.
>
> We take the first 500 data points that we already have (ETTh1, ETTh2, ETTm1, ETTm2) as a standalone small data set to evaluate our method under
> . The results (MSE) are shown in the following table and demonstrate that our KokerNet consistently outperforms the others.
>
> |                           | **ETTh1_500** | **ETTh2_500** | **ETTm1_500** | **ETTm2_500** |
> |---------------------------|---------------|---------------|---------------|---------------|
> | **DMD**                   | 3.0720        | 7.8345        | 1.2155        | 2.6091        |
> | **eDMD**                  | 3.8089        | 6.2291        | 1.0624        | 3.5871        |
> | **KRR[1]**   | 2.5595        | 13.3960       | 2.1210        | 7.2693        |
> | **PCR[1]**   | 3.9560        | 12.7278       | 3.2041        | 8.0007        |
> | **sKAF [1]**              | 2.3596        | 15.9180       | 0.8154        | 10.7259       |
> | **KokerNet**              | **0.7682**    | **0.5754**    | **0.3716**    | **0.5381**    |
>
> [1] Meanti, G., Chatalic, A., Kostic, V., Novelli, P., Pontil, M., & Rosasco, L. (2024). Estimating Koopman operators with sketching to provably learn large scale dynamical systems. Advances in Neural Information Processing Systems, 36.
>
> [2] Dimitrios Giannakis, Amelia Henriksen, Joel A. Tropp, and Rachel Ward. Learning to forecast dynamical systems from streaming data. SIAM Journal on Applied Dynamical Systems, 22(2): 527–558, 2023

---

### Official Review · Reviewer_u2dN · 2024-11-04

**Soundness:** 2
**Presentation:** 2
**Contribution:** 2
**Rating:** 5
**Confidence:** 4

**Summary:**

This paper builds on Koopman operator methodology to address non-stationary time series forecasting in an efficient way based on Random Fourier Features. The authors consider the Reproducing Hilbert space built from Fourier Features as the measurement function space, and report approximation results on Gram matrix as well on the Koopman operator eigenfunctions. In the section dedicated to learning, they assume a decomposition of the time series into a stationary and no stationary part and learn the corresponding global and locals Koopman operators following the steps of (Liu et al. 2023). A statistical test (KS) is proposed to measure the stationarity of the data from which helps to define the decomposition in stationary/non-stationary signals. Finally they introduce an alignment loss that measures how the forecasted distribution differs from the ground truth assuming Gaussianity  of the state variable distribution. Experimental results include a few competitors and feature different time-series benchmarks.

**Strengths:**

This paper tackles non-stationary times-series forecasting, a classical but still crucial in many applications. They propose to use the Koopman operator machinery to adress this issue. Leveraging random Fourier methodology for the measurement space, they can propose an efficient estimation of these operators that they learn in the context of a decomposiiton of the time-series into a stationary and non-stationary part.
The paper comes with a new (but relatively direct) result on the estimation of eigenfunctions of a Koopman operator when leveraging the finite dimensional reproducing Kernel Hilbert space induced by the Random Fourier Features.
A secondary contribution is the proposition to use a Sinkhorn loss to measure the distance between the distribution of forecasts and the ground truth.

**Weaknesses:**

While very promising this paper, this paper seems to suffer from incompleteness in its presentation and should be re-writtent to make it reproducible.

The learning of the global and local koopman operator is not clearly posed but are direclty inherited from (Liu et al 2023) : the loss function is even not mentioned, here likely the square loss given the equations. A constraint is evocated but never described in the context of learning: when is it used?

The decomposition into stationary and non-stationary parts which is crucial here is only discussed in the appendix, but it remains very unclear how the stationarity index is used (probably a typo on S_V / S_alpha  ?). I also find disturbing that the authors refer to a Python code and not mathematical equations.

Finally it is important to refer to previous papers on:
- Random Fourier Features for Koopman operator learning: Salam, T., Li, A. K., & Hsieh, M. A. (2023, June). Online Estimation of the Koopman Operator Using Fourier Features. In Learning for Dynamics and Control Conference (pp. 1271-1283). PMLR.
other attempts to work with finite dimensional measurement spaces
- (available in arxiv in 2023)
Meanti, G., Chatalic, A., Kostic, V., Novelli, P., Pontil, M., & Rosasco, L. (2024). Estimating Koopman operators with sketching to provably learn large scale dynamical systems. Advances in Neural Information Processing Systems, 36.

- in the experimental part, the authors refer to loss terms they never introduced - they also do not mention how they choose the number of random Fourier feature no provide an analysis of it.

**Questions:**

please could you:
(1) indicate in what extent the stationarity index is new ?(if not, please refer to the literature on KS test for stationarity measurement)
(2) explain in details how you use the stationarity index that you propose and finally choose the way you decompose the time-series.
most importantly:
(3) could you explain how you use the so-called constraint module and provide a complete algorithm.

---

> ### Author Response · Authors · 2024-11-18
> **Response**
>
> Thank you for taking the time to our work. Your valuable comments are helpful in improving our work. We will provide a point-to-point response in the rebuttal.
>
> **Q1: When is the constraint used?**
>
> **A1:**  In the following, we will provide the detailed process and show it in the following **Algorithm 1**.
>
> Specifically, for the stationary component $\bf{X}_ {\text{s}}$, we assume the distribution $\mathcal{N}(\bf{\mu}_ {\text{s}}, \bf{\delta}_ {\text{s}}^2)$ is constant. We just need to let the distribution of the forecasting $\hat{\bf{X}}_ {\text{s}}^{\text{fore}}$ closed to the real distribution $\mathcal{N}(\bf{\mu}_ {\text{s}}, \bf{\delta}_ {\text{s}}^2)$ by minimizing the distribution loss $\mathcal{L}_ {\text{dis}}^{\text{s}} = \mathcal{L}(\mathcal{N}(\bf{\mu}_ {\text{s}}, \bf{\delta}_ {\text{s}}^2), \mathcal{N}(\hat{\bf{\mu}}_ {\text{s}}, \hat{\bf{\delta}}_ {\text{s}}^2))$, where $\mathcal{N}(\hat{\bf{\mu}}_ {\text{s}}, \hat{\bf{\delta}}_ {\text{s}}^2)$ is the distribution of the forecasting $\hat{\bf{X}}_ {\text{s}}^{\text{fore}}$.
>
> For the non-stationary component $\bf{X}_ {\text{ns}}$, similar to the local Koopman operator learning in section 3.2, we explore the variation of the distribution by introducing the distribution Koopman operator $\mathcal{K}_ {\text{dis}}$ (Eq. (13)). In this section, we minimize the distribution loss $\mathcal{L}_ {\text{dis}}^{\text{ns}} = \mathcal{L}(\mathcal{N}(\bf{\mu}_ {\text{ns}}^i, \bf{\delta}_ {\text{ns}}^{2, i}), \mathcal{N}(\bf{\hat{\mu}}_ {\text{ns}}^i, \bf{\hat{\delta}}_ {\text{ns}}^{2,i})), i=2, \cdots, Q$, where $ \mathcal{N}(\bf{\mu}_ {\text{ns}}^i, \bf{\delta}_ {\text{ns}}^{2, i})$ is the ground truth, and $\mathcal{N}(\bf{\hat{\mu}}_ {\text{ns}}^i, \bf{\hat{\delta}}_ {\text{ns}}^{2,i})$ is the prediction based on the $\mathcal{K}_ {\text{dis}}$. Furthermore, we align the predicted distribution with the distribution of forecasts by minimizing the loss $\mathcal{L}_ {\text{dis}}^{\text{alig}} = \mathcal{L}(\mathcal{N}(\hat{\bf{\mu}}_ i, \hat{\bf{\delta}}_ i^2), \mathcal{N}(\bf{\hat{\mu}}_ {\text{ns}}^i, \bf{\hat{\delta}}_ {\text{ns}}^{2,i}))$.
>
> **Q2: Global and local Koopman operators and loss function**
>
> **A2:**  For the global Koopman operator, we set $\bf{X}_ {\text{s}} = [\bf{X}_ {\text{s}}^{\text{back}}, \bf{X}_ {\text{s}}^{\text{fore}}]$. We first encode $\bf{X}_ {\text{s}}^{\text{back}}$ into the feature space \emph{i.e., $\bf{Z}_ {\text{s}}^{\text{back}} =  g(\bf{X}_ {\text{s}}^{\text{back}})$}, and then forecast by the global Koopman operator $\bf{Z}_ {\text{s}}^{\text{fore}} = \mathcal{K}_ {\text{s}}\bf{Z}_ {\text{s}}^{\text{back}}$. Finally, we decode the forecasting by a decoder: $\hat{\bf{X}}_ {\text{s}}^{\text{fore}} = \Phi_ {\text{de}}(\bf{Z}_ {\text{s}}^{\text{fore}})$. In this framework, the operation $\bf{Z}_ {\text{s}}^{\text{fore}} = \mathcal{K}_ {\text{s}}\bf{Z}_ {\text{s}}^{\text{back}}$ is a linear mapping and is learned by minimizing the loss $\mathcal{L}_ {\text{fore}}^{\text{s}} = \ell(\bf{X}_ {\text{s}}^{\text{fore}}, \hat{\bf{X}}_ {\text{s}}^{\text{fore}})$.
>
> For the local Koopman operator, we first divide the non-stationary component $\bf{X}_ {\text{ns}}$ into $Q$ segments $\bf{X}_ {\text{ns}} = [\bf{x}_ 1, \bf{x}_ 2, \dots,  \bf{x}_ Q]$, and encode them into the feature space \emph{i.e., $\bf{z}_ i = g(\bf{x}_ i), i=1,\dots, Q$}. Then, we construct $\bf{Z}^{\text{back}} = [\bf{z}_ 1, \dots, \bf{z}_ {Q-1}]$ and $\bf{Z}^{\text{fore}} = [\bf{z}_ 2, \dots, \bf{z}_ {Q}]$, and compute $\mathcal{K}_ {\text{ns}} = (\bf{Z}^{\text{back}}) (\bf{Z}^{\text{fore}})^\top$. Moreover, the computed $\mathcal{K}_ {\text{ns}}$ is used to forecast $\hat{\bf{z}}_ i = \mathcal{K}_ {\text{ns}}\bf{z}_ {i-1}$, and a decoder is applied to decode the forecasting $\hat{\bf{x}}_ i = \Psi_ {\text{de}}(\hat{\bf{z}}_ i), i=2,\dots, Q$. In this process, $\mathcal{K}_ {\text{ns}}$ is learned by minimizing the loss $\mathcal{L}_ {\text{fore}}^{\text{ns}} = \sum_ {i=2}^Q \ell(\bf{x}_ i, \hat{\bf{x}}_ i)$.
>
> In our KokerNet, the loss function consists of three parts, including the forecasting loss $\mathcal{L}_ {\text{fore}} = \mathcal{L}_ {\text{fore}}^{\text{s}} + \mathcal{L}_ {\text{fore}}^{\text{ns}}$, the distribution loss $\mathcal{L}_ {\text{dis}} = \mathcal{L}_ {\text{dis}}^{\text{ns}} + \mathcal{L}_ {\text{dis}}^{\text{alig}}$, and the reconstruction loss $\mathcal{L}_ {\text{rec}} = \ell(\bf{X}_ T, \hat{\bf{X}}_ {\text{s}} + \hat{\bf{X}}_ {\text{ns}})$, $\hat{\bf{X}}_ {\text{s}} = [\Phi_ {\text{de}}(\bf{Z}_ {\text{s}}^{\text{back}}), \Phi_ {\text{de}}(\bf{Z}_ {\text{s}}^{\text{fore}})]), \hat{\bf{X}}_ {\text{ns}} = [\Psi_ {\text{de}}(\bf{z}_ 1), \cdots, \Psi_ {\text{de}}(\bf{z}_ Q)]$ are the reconstruction by the decoder for the stationary and non-stationary components, respectively.

---

> > ### Author Response · Authors · 2024-11-18
> > **Response**
> >
> > **Q3: The decomposition of the time series**
> >
> > **A3:** For the decomposition, we first remove its global trends and seasonal effects. Such operations would cause the residual of the time series to tend to be stochastic fluctuation, which is the main attribution of the non-stationarity of real-world time series. This step is conducted by the additive model of the seasonal decompose function within the Pytorch (in the following, we will show the detailed process mathematically).  Then, the index $S_v$ is calculated based on the Kolmogorov–Smirnov test to determine the proportion of the stationary and non-stationary components. After that, we perform the Fourier transform to calculate the frequency spectrum and sort all frequencies by the number of occurrences. Finally, the top $\alpha$ percent of the frequency spectrum are considered as the components of the stationary, while the remaining are considered as the components of the non-stationary. Note that the value of $\alpha$ is determined by the index $S_v$.
> >
> > Removing the global trends and seasonal effects mainly includes four steps:
> >
> > (1) **Determine the seasonal cycle (i.e., period) of the data**. The period denotes the length of the season.
> >
> > (2) **Compute the trend components**. The trend components are commonly calculated by the Moving Average way. Formulating as:
> > $$
> >         \bf{X}_ {\text{trend}}(t) = \frac{1}{p_ l} \sum_ {k=-\frac{(p_ l-1)}{2}}^{\frac{(p_ l-1)}{2}} \bf{X}(t+k)
> > $$
> > where $p_l$ is the length of the moving window.
> >
> > (3) **Compute the seasonal components**. The seasonal component is the remaining cyclical pattern after removing the trend component. Formulating as:
> > $$
> >         \bf{X}_ {\text{seasonal}}(t) = \frac{1}{M} \sum_{i=1}^{M} (\bf{X}(t + is_l) - \bf{X}_{\text{trend}}(t + is_l))
> > $$
> > where $s_l$ is the seasonal period, $M$ is the number of times the seasonal period repeats.
> >
> > (4) **Compute the residual** by $\bf{r} = \bf{X} - \bf{X}_ {\text{trend}} - \bf{X}_{\text{seasonal}}$.
> >
> > **Q4: The stationary index**
> >
> > **A4:**  In our paper, we define the index $S_v = p\overline{p}$, where $p$ and $\overline{p}$ denote the statistical magnitude of KS test in the time and frequency domain, respectively. This index measures the variation in both the time and frequency domains via multiplication. Especially for the frequency domain, it is inspired by the autocorrelation function, which can measure the relationship between a time series and its lagged versions. It shows that the stationarity is manifested in the uncertainty about the spectrum. Therefore, we take both the time and frequency domains into account to compute the stationary index.
> >
> > For a time series, we compute the stationary index $S_v$ based on the steps discussed above. One can set a threshold, if the index is less than this threshold, the time series is considered to be stationary. If the index is greater than this threshold, the time series is considered to be non-stationary and is decomposed into stationary and non-stationary components. Among them, the non-stationary component accounts for $S_v$, and the stationary component accounts for $1-S_v$. In our paper, $\alpha = 100 \times (1-S_v)$, and $S_\alpha$ is a set of top $\alpha$ percent of the frequency spectrum, *i.e.,* the set of the global shared frequency spectrum. The remaining corresponds to the non-stationary component. The threshold is a hyper-parameter, and we set it as 0.1 in our paper.
> >
> > **Q5: How do choose the number of random Fourier features?**
> >
> > **A5:** In **Theorem 3.1**, we show that when the sampling number $M \geq \frac{2\delta(3\sqrt{n}+2\Delta)}{3\Delta^2} \text{ln}\frac{8\sqrt{n}}{\rho}$, the kernel in Eq. (2) can be approximated by the kernel in Eq. (3). This provides a theoretical guide for us to choose the number of random features. From **Theorem 3.1**, we can observe that the larger the sampling number, the better the approximation effect theoretically. In the revised manuscript, we will discuss it in detail.

---

> > > ### Comment · Reviewer_u2dN · 2024-11-28
> > > **Feedback on the authors' answers**
> > >
> > > Thank you for your detailed feedback and numerous insights. I do appreciate the efforts.
> > > I still think the paper requires an important revision in its presentation and in the clarification of its claims compared to the existing literature about apporximating Koopman operators (all the DMD's recent papers).

---

> > ### Author Response · Authors · 2024-11-19
> > **Response**
> >
> > **Algorithm 1:** KokerNet for Time Series Forecasting
> >
> > **Input:**  $\bf{X}_T$ with $T$ time points.
> >
> > **Output:**  $\mathcal{K}_ {\text{s}}, \mathcal{K}_ {\text{ns}}, \mathcal{K}_ {\text{dis}}, g_ {\bf{\Theta}}(\cdot), \bf{\Theta} = \{\omega_1, \cdots, \omega_M\}, \Phi_{\text{de}}, \Psi_{\text{de}}$.
> >
> > ----
> > 1. Calculate $S_v \gets p\overline{p}$ based on Eq (7) and Eq (9).
> > 2. Divide $\bf{X}_ T$ into $\bf{X}_ {\text{s}}$ and $\bf{X}_ {\text{ns}}$ based on $S_v$.
> >
> >  3.  **Repeat:**
> >  4.  $\quad$  $\bf{X}_ {\text{s}} = [\bf{X}_ {\text{s}}^{\text{back}}, \bf{X}_ {\text{s}}^{\text{fore}}]$:
> > 5. $\quad$ Compute the distribution $\mathcal{N}(\bf{\mu}_ {\text{s}}, \bf{\delta}_ {\text{s}}^2) \gets \bf{X}_ {\text{s}}^{\text{back}}$;
> > 6. $\quad$ Compute $\bf{Z}_ {\text{s}}^{\text{back}} \gets g(\bf{X}_ {\text{s}}^{\text{back}})$, and forecast $\bf{Z}_ {\text{s}}^{\text{fore}} \gets \mathcal{K}_ {\text{s}} \bf{Z}_ {\text{s}}^{\text{back}}$ with $\mathcal{K}_ {\text{s}}$;
> > 7. $\quad$ Decode $\bf{Z}_ {\text{s}}^{\text{fore}}$ with the decoder $\Phi_ {\text{de}}$, $\hat{\bf{X}}_ {\text{s}}^{\text{fore}} \gets \Phi_ {\text{de}}(\bf{Z}_ {\text{s}}^{\text{fore}})$;
> > 8. $\quad$ Compute the distribution $\mathcal{N}(\hat{\bf{\mu}}_ {\text{s}}, \hat{\bf{\delta}}_ {\text{s}}^2) \gets \hat{\bf{X}}_ {\text{s}}^{\text{fore}}$;
> > 9. $\quad$ Compute the loss $\mathcal{L}_ {\text{fore}}^{\text{s}}$; $\mathcal{L}_ {\text{dis}}^{\text{s}}$.
> >
> > 10. $\quad$ $\bf{X}_ {\text{ns}} = [\bf{x}_1, \cdots, \bf{x}_Q]$:
> > 11. $\quad$ Compute the distribution $\mathcal{N}(\bf{\mu}_ {\text{s}}, \bf{\delta}_ {\text{s}}^2) \gets \bf{X}_ {\text{s}}^{\text{back}}$;
> >    12. $\quad$ Compute $\bf{Z}_ {\text{s}}^{\text{back}} \gets g(\bf{X}_ {\text{s}}^{\text{back}})$, and forecast $\bf{Z}_ {\text{s}}^{\text{fore}} \gets \mathcal{K}_ {\text{s}} \bf{Z}_ {\text{s}}^{\text{back}}$ with $\mathcal{K}_ {\text{s}}$;
> >    13. $\quad$ Decode $\bf{Z}_ {\text{s}}^{\text{fore}}$ with the decoder $\Phi_ {\text{de}}$, $\hat{\bf{X}}_ {\text{s}}^{\text{fore}} \gets \Phi_ {\text{de}}(\bf{Z}_ {\text{s}}^{\text{fore}})$;
> >    14. $\quad$ Compute the distribution $\mathcal{N}(\hat{\bf{\mu}}_ {\text{s}}, \hat{\bf{\delta}}_ {\text{s}}^2) \gets \hat{\bf{X}}_ {\text{s}}^{\text{fore}}$;
> >    15. $\quad$ Compute the loss $\mathcal{L}_ {\text{fore}}^{\text{s}}$; $\mathcal{L}_ {\text{dis}}^{\text{s}}$;
> > 16. $\quad$  Return: $\mathcal{N}(\bf{\hat{\mu}}_ {\text{ns}}^i, \bf{\hat{\delta}}_ {\text{ns}}^{2,i})$
> > 17.  $\quad$ $\bf{z}_i \gets g(\bf{x}_i), i = 1, \dots, Q$;
> >    18. $\quad$  $\bf{Z}^{\text{back}} \gets [\bf{z}_ 1, \dots, \bf{z}_ {Q-1}]$, $\bf{Z}^{\text{fore}} \gets [\bf{z}_ 2, \dots, \bf{z}_ {Q}]$;
> >    19. $\quad$ $\mathcal{K}_ {\text{ns}} \gets (\bf{Z}^{\text{back}}) (\bf{Z}^{\text{fore}})^\top$;
> >    20. $\quad$ $\hat{\bf{z}}_ i \gets \mathcal{K}_ {\text{ns}} \bf{z}_ {i-1}$;
> >    21. $\quad$ Decode $\hat{\bf{x}}_ i \gets \Psi(\hat{\bf{z}}_ i), i = 2, \dots, Q$;
> >    22. $\quad$ Compute the distribution $\mathcal{N}(\hat{\bf{\mu}_ i}, \hat{\bf{\delta}}_ i^2) \gets \hat{\bf{x}}_i, i = 2, \dots, Q$;
> >    23. $\quad$ Compute the loss $\mathcal{L}_ {\text{fore}}^{\text{ns}}$; $\mathcal{L}_ {\text{dis}}^{\text{alig}} \gets \mathcal{L}(\mathcal{N}(\hat{\bf{\mu}}_ i, \hat{\bf{\delta}}_ i^2), \mathcal{N}(\bf{\hat{\mu}}_ {\text{ns}}^i, \bf{\hat{\delta}}_ {\text{ns}}^{2,i})), i = 2, \dots, Q$.
> >
> > 24.  $\quad$ Compute the total loss:   $\mathcal{L}_ {\text{KokerNet}} \gets \mathcal{L}_ {\text{fore}}^{\text{s}} + \mathcal{L}_ {\text{dis}}^{\text{s}} + \mathcal{L}_ {\text{fore}}^{\text{ns}} + \mathcal{L}_ {\text{dis}}^{\text{ns}} + \mathcal{L}_ {\text{dis}}^{\text{alig}} + \mathcal{L}_ {\text{rec}}$.
> >
> > 25. $\quad$ **Update**
> >
> > 26. **Until:** Convergence

---

> ### Author Response · Authors · 2024-11-18
> **Response**
>
> **Q6: Previous papers**
>
> **A6:** Thank you for providing us with references that let us know more about the work of learning koopman operators based on kernel methods. We have already read the two of papers listed. The main idea of these two papers and our work is incorporating the kernel method in the Koopman theory and further learning the Kppman operator in a finite-dimensional measurement space by kernel approximation.
>
> "Estimating Koopman operators with sketching to provably learn large-scale dynamical systems" approximates the kernel using the Nystrom method, which depends very much on the selection of induction points and does not take all the data into account. Compared to the Nystrom method, the random Fourier feature (RFF) can make a "good" approximation of a kernel using all the data. Both "Online Estimation of the Koopman Operator Using Fourier Features" and our paper utilize RFF to approximate the kernel and construct a finite-dimensional measurement space. The main difference between our work and this one is that the learning of Koopman operator is realized through the learning of the kernel network in our work. Concretely, the Koopman operator learning can be formulated as: $\hat{\bf{x}} = \Phi_ {de}(\mathcal{K}g(\bf{x}))$, where $\bf{x}$ and $\hat{\bf{x}}$ are the input and output, respectively. $g$ denotes the kernel mapping with RFF, $\mathcal{K}$ denotes the Koopman operator, and $\Phi_ {de}$ denotes a decoder. We make $\mathcal{K}(\cdot)$ as the linear mapping, and thus $\mathcal{K}g(\bf{x})$ can be considered as a kernel network, stacking the non-linear kernel mapping with linear mapping.
>
> **If you have any further questions, we will discuss them in a timely manner. That could help improve our work.**

---

> > ### Comment · Reviewer_u2dN · 2024-11-28
> > **Other Feedback**
> >
> > Hi,
> > about the 2 references on kernel approximation used for Koopman operator (Meanti et al.) and (Salam et al.) cited in my review, an interesting addition to your comparison table.
> > thank you

---

> > > ### Author Response · Authors · 2024-11-30
> > > **Response**
> > >
> > > Thank you for taking the time to read our rebuttal carefully and reply. We are also pleased to have received positive feedback. To further evaluate the performance of our proposal, we compare our KokerNet with the widely used DMD and eDMD methods. The code of DMD and eDMD is from the public PyDMD codebase in GitHub. We also compare with kernel approximation used for the Koopman operator (Meanti et al.) (KRR and PCR) cited in your review.  Due to the lack of released sources and insufficient time for reproduction, we can only present the comparison results in the following table. More DMD-based methods will be included in the revised manuscript.
> > >
> > > Since DMD-based methods need more time to run, we take the first 500 data points that we already have (ETTh1, ETTh2, ETTm1, ETTm2) as a standalone small data set to evaluate our method under $H=48$. The results (MSE) are shown in the following table and demonstrate that our KokerNet consistently outperforms the others.
> > >
> > > |                           | **ETTh1_500** | **ETTh2_500** | **ETTm1_500** | **ETTm2_500** |
> > > |---------------------------|---------------|---------------|---------------|---------------|
> > > | **DMD**                   | 3.0720        | 7.8345        | 1.2155        | 2.6091        |
> > > | **eDMD**                  | 3.8089        | 6.2291        | 1.0624        | 3.5871        |
> > > | **Meanti et al. (KRR)**   | 2.5595        | 13.3960       | 2.1210        | 7.2693        |
> > > | **Meanti et al. (PCR)**   | 3.9560        | 12.7278       | 3.2041        | 8.0007        |
> > > | **sKAF [1]**              | 2.3596        | 15.9180       | 0.8154        | 10.7259       |
> > > | **KokerNet**              | **0.7682**    | **0.5754**    | **0.3716**    | **0.5381**    |
> > >
> > >
> > > [1] Dimitrios Giannakis, Amelia Henriksen, Joel A. Tropp, and Rachel Ward. Learning to forecast
> > > dynamical systems from streaming data. SIAM Journal on Applied Dynamical Systems, 22(2):
> > > 527–558, 2023

---

> > > > ### Author Response · Authors · 2024-12-02
> > > > **Response**
> > > >
> > > > Dear Reviewer u2dN,
> > > >
> > > > Thank you very much for improving the score and apologize for our continuous messages. Since the end of the discussion period is approaching, we would appreciate if you could let us know whether the above answer and additional experiment address your comments or not and let us know if you have additional questions, comments, and concerns related to these points. Thanks again.

---

### Author Response · Authors · 2024-11-24
**Response**

Dear Reviewers,

We sincerely appreciate your time and effort in reviewing our work and providing insightful comments, which could help improve our work.

We provided a point-to-point response to your concerns, and we hope our responses have adequately addressed your comments. As the author-reviewer discussion phase is drawing to a close, we would like to confirm whether our responses have effectively addressed your concerns. If you require further clarification or have any additional concerns, please do not hesitate to reach out. We are more than willing to discuss them with you on time.

Best regards,

---

### Meta-Review · Area_Chair_op8m · 2024-12-18

**Metareview:**

**(a) Summary**

This paper investigates a Koopman operator-based approach for time series forecasting. The proposed method constructs a measurement function space by approximating the kernel function using a low-dimensional feature mapping derived from Bochner’s theorem. The key technical contribution lies in decomposing a given time series into stationary and non-stationary components using the KS test. Global and local Koopman operators are then learned to model the dynamics of these components effectively. The proposed method, called KokerNet, is evaluated empirically on real-world datasets.

**(b) Strengths**

- **Technical quality:** The proposed decomposition of time series into stationary and non-stationary components using the KS test appears to be both effective and practical.
- **Approach:** Combining spectral kernel methods with Koopman operator theory is a potentially impactful technical contribution.

**(c) Weaknesses**

Several critical concerns raised by reviewers include:

- **Presentation and Novelty:** The presentation requires significant improvement, which requires a major revision. It is unclear which parts of the method are novel contributions and which parts follow existing literature. This lack of clarity deteriorates the overall quality and novelty of the paper, as highlighted by **Reviewer 4P4B**. Additionally, the decomposition process, which is a crucial aspect of the method, is insufficiently explained, as noted by **Reviewer u2dN**.
- **Related work and empirical evaluation:** The discussion and empirical comparison with related work are insufficient. In particular, while I understand that the proposed approach is different from the DMD approach, it is essential to carefully compare and discuss these approaches within the context of time series forecasting as suggested by **Reviewer 4P4B**.

**(d) Decision Reasoning**

As noted above, significant issues with the paper’s presentation remain, even after the authors’ rebuttal. These concerns must be addressed through at least one more round of revisions before the paper can be considered for publication. Consequently, I recommend rejecting the paper at this stage. However, I believe that a revised version addressing these issues could make a meaningful contribution to the community.

**Additional Comments On Reviewer Discussion:**

Crucial concerns about the presentation and novelty have been raised by the reviewers. While the authors have made significant efforts in their rebuttal and partially addressed these issues with a revised version of the paper, it is still not ready for publication.

---

### Decision · Program_Chairs · 2025-01-22

Reject